# Learning to Partially Defer for Sequences

## Abstract

In the Learning to Defer (L2D) framework, a prediction model can either make a prediction or defer it to an expert, as determined by a rejector. Current L2D methods train the rejector to decide whether to reject the *entire prediction*, which is not desirable when the model predicts long sequences. We present an L2D setting for sequence outputs where the system can defer *specific outputs* of the whole model prediction to an expert in an effort to interleave the expert and machine throughout the prediction. We propose two types of model-based post-hoc rejectors for pre-trained predictors: a token-level rejector, which defers specific token predictions to experts with next token prediction capabilities, and a one-time rejector for experts without such abilities, which defers the remaining sequence from a specific point onward. In the experiments, we also empirically demonstrate that such granular deferrals achieve better cost-accuracy tradeoffs than whole deferrals on Traveling salesman solvers, News summarization, and Weather prediction.

## 1 Introduction

Trustworthiness of AI is under scrutiny with its increased deployment in highly consequential areas like healthcare (Ker et al., 2017; Courtiol et al., 2019; Asan et al., 2020) and criminal justice (Dressel & Farid, 2018; Rigano, 2019; Alikhademi et al., 2022). Hybrid intelligent systems (Kamar, 2016; Dellermann et al., 2019; Akata et al., 2020; Maadi et al., 2021) approach this problem through collaborations between either humans and machines or two machines to improve confidence in the system. Specifically, Learning to Defer (L2D) (Madras et al., 2018) accommodates these teams by allowing the model to defer to an expert when it is uncertain about a prediction task with the objective of maximizing overall system accuracy while minimizing deferral costs.

Existing L2D methods focus on multiclass (Mozannar & Sontag, 2020; Cao et al., 2022; Verma & Nalisnick, 2022; Mao et al., 2024a;c) and scalar regression (Cheng et al., 2024; Mao et al., 2024b) problems. However, these methods only support complete deferral of predictions, resulting in costs that scale with the output size, when applied to large *sequence outputs* (Narasimhan et al., 2024). This is inefficient if only parts of the model's prediction are inaccurate. The prevalence of sequences in areas such as drug discovery (Wang et al., 2018; Jumper et al., 2021; Dauparas et al., 2022; Nijkamp et al., 2023) and language modeling (Kenton & Toutanova, 2019; Brown et al., 2020; Raffel et al., 2020) calls for a cost-effective extension of the L2D framework to handle sequential outputs.

To this end, we propose a fine-grained approach to deferral where the system has the flexibility to defer *portions* of the predicted sequence. For example, in part of speech tagging, this mechanism would only require the expert to tag a few uncertain sentences in a paragraph rather than the whole paragraph, reducing the time cost which increases with the number of words to be tagged. This can, therefore, achieve a more favorable cost-accuracy tradeoff, which can further improve as the deferrals become more precise - for example, at the word-level. Additionally, parts of the prediction task completed by the predictor can also function as contextual clues for the language expert to fill in the deferred parts, further minimizing deferral costs.

While partial deferral is therefore desirable, the granularity of such deferrals depends on the size of the sequence segments the expert can predict at a time. For instance, if the expert is a next-token predictor, it can complete spans as small as a single token—*token-level* deferral. This is especially valuable when

the model's predictive quality across all parts of the output are dependent. Specifically, in autoregressive prediction, uncertainty at earlier prediction points can propagate and degrade later points (Bengio et al., 2015; Lamb et al., 2016; Schmidt, 2019). Thus, to achieve a better cost-accuracy tradeoff, the token-level rejector can orchestrate a dynamic collaboration between the predictor and the expert where the predictor receives an expert prediction on an early substructure before the model veers "off-course".

This, however, is only feasible with experts that can make predictions one token at a time. That said, partial deferrals can still be implemented as long as the expert is at least capable of completing the predicted sequence given a partial subsequence from the model. We refer to this type of deferral as *one-time* deferral. For example, Gurobi solvers for Traveling salesman problems lack the ability to iteratively select the next city in the tour, but they can use partial subtours generated by models as constraints to generate the entire tour at a reduced cost. In this case, the one-time rejector would select a point in the sequence from which the expert completes the sequence.

In this paper, we make the following contributions:

- We present a novel L2D framework where the rejector interweaves the expert's and machine's predictions to improve cost-accuracy tradeoffs. We divide this setting into token-level and one-time deferrals based on the granularity of the deferrals, thereby accommodating various types of experts.

- We propose convex surrogate losses that offer Bayes consistency guarantees leading to implementable model-based post hoc rejectors for pre-trained predictors for each type of deferral. We further analyze the surrogates theoretically through generalization bounds.

- We empirically demonstrate improved cost-accuracy tradeoffs when using model-based rejectors for either deferrals against whole sequence rejectors with Traveling salesman solvers, News summarization, and Weather prediction.

## 2 Background

### 2.1 Learning to Defer

Deferrals, also known as rejections or abstentions, are classified into two types - *confidence*-based (Chow, 1957; Bartlett & Wegkamp, 2008; Yuan & Wegkamp, 2010; Ramaswamy et al., 2018) and *model*-based (Cortes et al., 2016; Madras et al., 2018; Mozannar & Sontag, 2020; Verma & Nalisnick, 2022; Cao et al., 2022; Mao et al., 2024a; Cheng et al., 2024; Mao et al., 2024b). *Confidence*-based methods learn a predictor that outputs confidences; a prediction is rejected whenever its confidence falls below a cost-dependent threshold. However, Madras et al. (2018) and Cortes et al. (2016) characterized the suboptimality of these methods and advocated for the rejector to be a model trained either simultaneously with the predictor or post-hoc: this is the *model*-based approach. This system seeks to minimize the following loss function:

$$\mathcal{L}_{\text{L2D}}\left(h, r, e, x, y\right) = l(y, h(x))\mathbf{1}_{r(x)<0} + c(x, y, e)\mathbf{1}_{r(x)\geq0} \tag{1}$$

where $h$ is the predictor, $r$ is the rejector, $e$ is the expert, $l(y, h(x))$ is the model loss, and $c(x, y, e)$ is the cost incurred when deferring to an expert.

Although our setting assumes access to a single expert, one-time deferral closely resembles L2D with multiple experts (Verma et al., 2023; Mao et al., 2024b). We formalize this similarity and its implications in Section 4.1. For $E$ experts, L2D minimizes:

$$\mathcal{L}_{\text{L2DMulti}}\left(h, r, x, y, \{e_j\}_{j=1}^E\right) = l(y, h(x))\mathbf{1}_{r(x)=0} + \sum_{j=1}^{E} c_j(x, y, e_j)\mathbf{1}_{r(x)=j} \tag{2}$$

where $c_j(x, y, e_j)$ is the cost of querying the $j^{\text{th}}$ expert, $e_j$. The rejector must decide which expert to defer to.

## 2.2 Consistent Losses

Direct minimization of Equations (1) and (2) is challenging due to the non-convexity and discontinuities. This can be addressed by optimizing a convex surrogate loss function which, upon minimization, will lead to the same outcome as minimizing the original loss. This property is formally called *Bayes Consistency*.

**Definition 2.1** (Consistency). Let the risk of a hypothesis $f$ with respect to a loss $l$ be $\mathcal{R}_l(f) = \mathbb{E}_{(x,y)\sim\mathcal{P}_{\mathcal{X},\mathcal{Y}}}[l(f(x),y)]$. Let the optimal risk with respect to $l$ out of all hypotheses in $\mathcal{F}$ be $\mathcal{R}_l^*(\mathcal{F}) = \inf_{f\in\mathcal{F}} \mathcal{R}_l(f)$. A surrogate loss function, $\phi(\cdot)$, is said to be $\mathcal{F}$-consistent with respect to $l$ if for any $f_n \in \mathcal{F}$, the following property holds true:

$$\lim_{n\to\infty} \mathcal{R}_\phi(f_n) - \mathcal{R}_\phi^*(\mathcal{F}) = 0 \Rightarrow \lim_{n\to\infty} \mathcal{R}_l(f_n) - \mathcal{R}_l^*(\mathcal{F}) = 0$$

When $\mathcal{F}$ is the class of all measurable functions, a surrogate $\phi$ is Bayes consistent for a target loss $l$ if there exists a nondecreasing function $\Gamma : \mathbb{R} \to \mathbb{R}$ continuous at 0 with $\Gamma(0) = 0$, such that for all $f$,

$$\mathcal{R}_l(f) - \mathcal{R}_l^* \leq \Gamma\left(\mathcal{R}_\phi(f) - \mathcal{R}_\phi^*\right)$$

where $\mathcal{R}_l^* = \inf_f \mathcal{R}_l(f)$ is the optimal risk over all measurable functions. We introduce surrogates for both token-level and one-time deferral (similar to Equations (1) and (2)) and prove they satisfy the above inequality, thereby establishing Bayes consistency.

# 3 Token-level partial deferral

Let $\mathcal{Y} \subseteq \bigcup_{\ell=1}^{L} \mathcal{V}^\ell$ denote the space of sequences over label set $\mathcal{V}$ with maximum length $L$. For $y \in \mathcal{Y}$, the $j^{\text{th}}$ token is $y_j \in \mathcal{V}$. $\mathcal{V}$ can be a finite vocabulary in the case of text sequences and be $\mathbb{R}$ when the output is a scalar time series. Let $\mathcal{X}$ be the feature space. In sequence generation, $\mathcal{X} \subseteq \mathbb{R}^d$ would be a $d$-dimensional feature space, whereas $\mathcal{X} = \mathcal{V}'^{L'}$ would be a sequence space in sequence-to-sequence learning. Let $\mathcal{P}_{X,Y}$ denote the data-generating distribution over $\mathcal{X} \times \mathcal{Y}$.

Suppose we have a predictor $h$ and a stronger expert $e$. $e$ takes in $x \in \mathcal{X}$ and the leftward context $\widehat{y}_{<j} = (\widehat{y}_1, \cdots, \widehat{y}_{j-1}) \in \mathcal{V}^{j-1}$ to predict $y_j$; repeated calls can auto-regressively complete the sequence. $e$ can, for instance, be a human or a large autoregressive model. However, querying $e$ is more expensive than calls to $h$. We, therefore, wish to design an L2D system where experts can complete uncertain tokens in the model's prediction, facilitating more fine-grained deferrals – referred to as *token-level* deferrals – yielding a more favorable cost-quality tradeoff than deferring the entire sequence.

If $h$ is autoregressive, token-level deferrals allows $h$ to leverage the expert-supplied tokens when generating subsequent tokens, enabling a seamless and adaptive collaboration. This mitigates uncertainty propagation, reduces the number of expert queries, and improves cost–effectiveness. While this paradigm can apply to all sequence predictors, we will specifically focus on a dynamic L2D system with an autoregressive predictor.

In this setting, at step $j$, the rejector $r_j$ takes in $(x, \widehat{y}_{<j})$ where $\widehat{y}_{<j}$ may mix predictor and expert tokens. When $r_j(x, \widehat{y}_{<j}) \geq 0$, it rejects the $j^{\text{th}}$ label prediction from $h$ and receives an expert prediction $\widehat{y}_j^e$. Otherwise, the model will keep the prediction from $h$, i.e. $\widehat{y}_j^h = h(x, \widehat{y}_{<j})$. For a fixed $h$ and a specific instance of $(x, \widehat{y}_{<j}, y)$, $r_j$ is trained to minimize:

$$\mathcal{L}_j\left(h, r_j, x, \widehat{y}_{<j}, y\right) = l\left(y, \widehat{y}_j^h\right) \mathbf{1}_{r_j\left(x, \widehat{y}_{<j}\right)<0} + c_j(x, \widehat{y}_{<j}, y)\mathbf{1}_{r_j\left(x, \widehat{y}_{<j}\right)\geq 0} \tag{3}$$

where $l(y, \widehat{y}_j^h)$ is the loss from using $\widehat{y}_j^h$ and $c_j(x, \widehat{y}_{<j}, y)$ is the deferral cost. $l(y, \widehat{y}_j^h)$ can be the mean squared error defined by $(y_j - \widehat{y}_j^h)^2$ or a 0-1 loss between the true $j^{\text{th}}$ token and $\widehat{y}_j^h$ in text sequences. Generally, $l\left(y, \widehat{y}_j^h\right)$ can be viewed as token-level feedback to generate an accurate complete sequence. $c_j(x, \widehat{y}_{<j}, y)$ can be a constant $c_j$ to reflect expert query costs but it can also be $l(y, \widehat{y}_j^e)$ to account for the expert's errors.

Finally, the predicted $j^{\text{th}}$ label $\widehat{y}_j = \widehat{y}_j^h \mathbf{1}_{r_j\left(x, \widehat{y}_{<j}\right)<0} + \widehat{y}_j^e \mathbf{1}_{r_j\left(x, \widehat{y}_{<j}\right)\geq 0}$ is appended to $\widehat{y}_{<j}$ to serve as input for $(j+1)^{\text{th}}$ token prediction, and this continues until $L$ labels are generated. The total loss for the sequence

is $\mathcal{L}^{\text{Token}}(h, r, x, y) = \frac{1}{L} \sum_{j=1}^{L} \mathcal{L}_j (h, r_j, x, \widehat{y}_{<j}, y)$ where $r = \begin{bmatrix} r_1 & \dots & r_L \end{bmatrix}^T$. The $\mathcal{L}^{\text{Token}}$-risk is denoted by $\mathcal{R}_{\text{Token}}(h, r) = \mathbb{E}_{(x,y) \sim \mathcal{P}_{X,Y}}[\mathcal{L}^{\text{Token}}(h, r, x, y)]$.

For analysis, we assume teacher-forced training (Williams & Zipser, 1989; Lamb et al., 2016): at $j^{\text{th}}$ step, $\mathcal{L}_j$ is evaluated on the output of $r_j$, but we roll out the leftward context for step $j+1$ using the oracle rejection decision $\mathbf{1}_{l(y, \widehat{y}_j^h) \geq c_j(x, \widehat{y}_{<j}, y)}$. This decouples the training distribution of $\widehat{y}_{<j}$ from the router's predictions; the context depends only on $(x, y)$ via the predictor and expert outputs. The theoretical guarantees established in this section apply to such teacher-forced inputs. In contrast, our strongest empirical results are obtained without teacher forcing to mitigate exposure bias; see Appendix H. Extending this theoretical analysis to training regimes without teacher-forcing remains an important direction for future work.

## 3.1 Surrogate Loss

As described in Section 2.2, minimizing the deferral loss directly is computationally difficult, so identifying an optimal rejector requires optimizing a convex surrogate loss that is Bayes consistent with respect to $\mathcal{L}^{\text{Token}}$. To this end, we propose the following surrogate loss function: $\mathcal{L}^{\phi}(h, r, x, y) = \frac{1}{L} \sum_{j=1}^{L} \mathcal{L}_j^{\phi}(h, r_j, x, \widehat{y}_{<j}, y)$ where

$$\mathcal{L}_j^{\phi}(h, r_j, x, \widehat{y}_{<j}, y) = l\left(y, \widehat{y}_j^h\right) \phi(r_j(x, \widehat{y}_{<j})) + c_j(x, \widehat{y}_{<j}, y)\phi(-r_j(x, \widehat{y}_{<j})) \tag{4}$$

and $\phi$ is a convex binary surrogate loss. $\phi$ could, for example, be the logistic loss, i.e., $\phi(z) = \log(1 + \exp\{-z\})$, or the square loss, i.e., $\phi(z) = (1 - z)^2$. $\mathcal{L}^{\phi}$-risk is characterized by $\mathcal{R}_{\phi}(h, r) = \mathbb{E}_{(x,y) \sim \mathcal{P}_{X,Y}}[\mathcal{L}^{\phi}(h, r, x, y)]$.

The optimization of $\mathcal{L}^{\phi}(h, r, x, y)$ is feasible due its convexity. Additionally, it upper bounds $\mathcal{L}^{\text{Token}}$ upto a scaling factor, providing alignment between reducing the surrogate loss and reducing the desired loss function. The proof of these properties can be found in Appendix A.1.

To show Bayes consistency of the surrogate, we relate the excess risk using $\mathcal{L}^{\text{Token}}$ and the excess risk using $\mathcal{L}^{\phi}$ in Lemma 3.1. The proof employs a specific type of distribution on $\mathcal{X} \times \{-1, 1\}$ with mass on functions of the conditional expectations of $l\left(y, \widehat{y}_j^h\right)$ and $c_j(x, \widehat{y}_{<j}, y)$ to relate the two inequalities - a strategy inspired by Mao et al. (2024a). The complete proof can be found in Appendix A.2.

**Lemma 3.1.** *Let $0 < \bar{c} \leq c_j(x, \widehat{y}_{<j}, y) \leq \bar{C} < \infty$ for all $j$, $0 \leq l(y, \widehat{y}_j^h) \leq \bar{l} < \infty$, and $\phi$ be a binary surrogate loss that satisfies the following inequality for any distribution over $\mathcal{X} \times \{-1, 1\}$ and any measurable function $f$:*

$$\mathcal{R}_{binary\ 0-1}(f) - \mathcal{R}_{binary\ 0-1}^* \leq \Gamma\left(\mathcal{R}_{binary\ \phi}(f) - \mathcal{R}_{binary\ \phi}^*\right)$$

*where $\mathcal{R}_{binary\ 0-1}$ is the binary 0-1 risk and $\Gamma : \mathbb{R}^+ \to \mathbb{R}^+$ is a non-decreasing concave function. Then, for any measurable $r$ and any $h$, the following inequality holds for any distribution over $\mathcal{X} \times \mathcal{Y}$:*

$$\mathcal{R}_{Token}(r, h) - \mathcal{R}_{Token}^*(h) \leq \tilde{\Gamma}\left(\mathcal{R}_{\phi}(r, h) - \mathcal{R}_{\phi}^*(h)\right)$$

*where $\tilde{\Gamma}(z) = \left(\bar{l} + \bar{C}\right)\Gamma\left(\frac{z}{\bar{c}}\right)$, $\mathcal{R}_{Token}^*(h) := \inf_r \mathcal{R}_{Token}(r, h)$, and $\mathcal{R}_{\phi}^*(h) := \inf_r \mathcal{R}_{\phi}(r, h)$*

Taking the limit on both sides of the inequality result of Lemma 3.1 proves consistency of the surrogate loss, formally stated in Theorem 3.2. The proof of Theorem 3.2 hinges on the fact that a classification-calibrated surrogate loss satisfies the condition presented in Lemma 3.1. A loss $\phi$ is considered binary classification calibrated if the minimizer of the pointwise $\phi$−risk or $\mathbb{E}_{y \sim \mathcal{P}_{\mathcal{Y}|\mathcal{X}=x}}[\phi(yf(x))]$ always has the same sign as $\mathbb{P}[Y = 1 \mid X = x] - \frac{1}{2}$. We defer the full proof of Theorem 3.2 and the formal definition of classification calibrated loss to Appendix A.3.

**Theorem 3.2** (Consistency). *If $0 < \bar{c} \leq c_j(x, \widehat{y}_{<j}, y) \leq \bar{C} < \infty$ for all $j$, $0 \leq l(y, \widehat{y}_j^h) \leq \bar{l} < \infty$, and $\phi$ be binary classification calibrated, then, for a fixed $h$, $\mathcal{L}^{\phi}$ is a Bayes consistent surrogate for $\mathcal{L}^{Token}$.*

## 3.2 Generalization Bounds

While Bayes consistency is desirable, the property only holds in the limit of infinite data and does not possess finite sample guarantees. In practice, finding a risk minimizer over the entire class of measurable functions

is extremely challenging, particularly because $\mathcal{P}_{X,Y}$ is usually unknown. With access to a rejector training data set $\mathcal{D}$ with $n$ samples, we can only learn the optimal $\widehat{r}_n \in \arg\min_{r \in \mathcal{F}} \widehat{\mathcal{R}}_\phi(r)$ from a restricted hypothesis class $\mathcal{F}$ by minimizing the empirical risk with respect to $\mathcal{L}^\phi$ - $\widehat{\mathcal{R}}_\phi(r) = \frac{1}{n} \sum_{(x^i, y^i) \in \mathcal{D}} \mathcal{L}^\phi(h, r, x^i, y^i)$. We now present an upper bound for the estimation error to characterize the quality of $\widehat{r}_n$ with respect to $\mathcal{F}$. Please refer to Appendix A.4 for the full proof.

**Theorem 3.3.** *Suppose $r \in \mathcal{F}$ which is composed of $\{\mathcal{F}_j\}_{j=1}^L$ such that $r_j \in \mathcal{F}_j$, $c_j(x) \leq \bar{C}$, and $l(y, \widehat{y}_j^h) \leq \bar{l}$. If $\phi$ is $\rho$-lipschitz continuous, then with probability $1 - \delta$, the following upper bound holds for the empirical risk minimizer $\widehat{r}_n$ with respect to $\mathcal{L}_\phi$:*

$$\mathcal{R}_\phi(\widehat{r}_n) - \mathcal{R}_\phi^*(\mathcal{F}) \leq \frac{4\rho\sqrt{2\left(\bar{C}^2 + \bar{l}^2\right)}}{L} \sum_{j=1}^L \mathfrak{R}_n(\mathcal{F}_j) + (\bar{C} + \bar{l})\sqrt{\frac{2\log\left(\frac{2}{\delta}\right)}{n}}$$

*where $\mathcal{R}_\phi^*(\mathcal{F}) = \inf_{r \in \mathcal{F}} \mathcal{R}_\phi(r)$ and $\mathfrak{R}_n(\mathcal{F}_j)$ is the Rademacher Complexity of $\mathcal{F}_j$*

From Theorem 3.3, the estimation error of using $\widehat{r}_n$ with respect to $\mathcal{L}^\phi$ depends on the Rademacher complexity $\mathfrak{R}_n(\mathcal{F})$ of the hypothesis class. Theorem 3.3 alone does not describe the excess risk with respect to the original token loss $\mathcal{L}^{\text{Token}}$. Using Lemma 3.1, we express the upper bound of the excess risk in Corollary 3.4, and the proof can be found in Appendix A.5.

**Corollary 3.4.** *Suppose $r \in \mathcal{F}$ which is composed of $\{\mathcal{F}_j\}_{j=1}^L$ such that $r_j \in \mathcal{F}_j$, $0 < \bar{c} \leq c_j(x) \leq \bar{C}$, and $l(y, \widehat{y}_j^h) \leq \bar{l} < \infty$. If $\phi$ is a $\rho$-lipschitz continuous binary classification-calibrated function, then with probability $1 - \delta$, the following upper bound on the excess risk with respect to $\mathcal{L}^{Token}$ holds for the empirical risk minimizer $\widehat{r}_n$ with respect to $\mathcal{L}_\phi$ for a fixed $h$:*

$$\mathcal{R}_{Token}(\widehat{r}_n, h) - \mathcal{R}_{Token}^*(h) \leq \tilde{\Gamma}\left(\mathcal{B}(\mathcal{F}) + \mathcal{A}_\phi(\mathcal{F}, h)\right)$$

*where $\tilde{\Gamma}$ is a non-decreasing function, $\mathcal{B}(\mathcal{F}) = \frac{4\rho\sqrt{2(\bar{C}^2 + \bar{l}^2)}}{L} \sum_{j=1}^L \mathfrak{R}_n(\mathcal{F}_j) + (\bar{C} + \bar{l})\sqrt{\frac{2\log(\frac{2}{\delta})}{n}}$, and $\mathcal{A}_\phi(\mathcal{F}, h) = \mathcal{R}_\phi^*(\mathcal{F}, h) - \mathcal{R}_\phi^*(h)$ is the approximation error.*

## 4 One-time partial deferral

Token-level deferrals are ideal to produce partial deferrals, but they hinge on the expert's capability to perform next-token prediction instead of generating the entire sequence at once. If the expert is at least capable of completing sequences using predicted subsequences from models, the system can achieve better cost-quality tradeoffs with partial deferrals compared to whole deferrals, albeit constrained by the lack of granularity in deferral, through *one-time* deferral. In the one-time partial deferral setting, the rejector takes in $x$ as input and identifies a point in sequence - referred to as the *deferral point* - from which the expert predicts the remaining sequence. When $r(x) = j$, the rejector chooses to reject the prediction of the sequence from the $j^{\text{th}}$ token onward and thereafter defers to the expert. In doing so, the system incurs a loss of $l(y, \widehat{y}_{<j})$ for using the first $j$ tokens from the predictor $h$ and pays a cost of $\tilde{c}_j(x, \widehat{y}_{<j}, y)$ for using the expert with the partial feedback from $h$. When $r(x) = 1$, the entire prediction task is deferred to the expert. If $r(x) = L + 1$, the model completes the full prediction without paying any cost for an expert. For this deferral point classification, our objective is to minimize the following loss function:

$$\mathcal{L}_{\text{Full}}^{\text{OneTime}}(h, r, x, y) = \sum_{j=1}^{L+1} \left(l(y, \widehat{y}_{<j}) + \tilde{c}_j(x, \widehat{y}_{<j}, y)\right) \mathbf{1}_{r(x)=j} \tag{5}$$

where $l(y, \widehat{y}_{<1}) = 0$

### 4.1 Influence of large $L$

This paradigm closely parallels learning to defer with multiple experts as their objectives are analogous (Equations (2) and (5)). Each position $j$ can be viewed as an "expert" whose prediction concatenates the

model prefix $\widehat{y}_{<j}$ with the expert-completed suffix $\widehat{y}_{\geq j}^e$. Prior work shows diminishing accuracy gains as the number of experts increases, which can reduce cost–effectiveness (Verma et al., 2023; Hemmer et al., 2023). This suggests that allowing deferral at every position may be over-resolved, since adjacent deferral points can yield similar accuracy.

Towards this end, we wish to allow for flexibility in adjusting the number of "experts". We achieve this by considering a generalization of one-time deferral where the predictor can defer to the expert on a select set of token positions specified by $\mathcal{J}$, instead of having all token positions be potential deferral points. For this generalization, we consider the following loss:

$$\mathcal{L}^{\text{OneTime}}(h, r, x, y) = \sum_{j \in \mathcal{J}} \left( l(y, \widehat{y}_{<j}) + \tilde{c}_j(x, \widehat{y}_{<j}, y) \right) \mathbf{1}_{r(x)=j} \tag{6}$$

The $\mathcal{L}^{\text{OneTime}}$-risk is denoted by $\mathcal{R}_{\text{OneTime}}(h, r) = \mathbb{E}_{(x,y) \sim \mathcal{P}_{X,Y}} [\mathcal{L}^{\text{OneTime}}(h, r, x, y)]$.

## 4.2 Surrogate loss

Let $\boldsymbol{g}$ consist of $|\mathcal{J}|$ scoring functions $g_j : \mathcal{X} \to \mathbb{R}$ where $j$ indexes the token positions in a sequence and let $r(x) = \arg\max_{j \in \mathcal{J}} g_j(x)$. We propose the following surrogate loss function:

$$\mathcal{L}^{\psi}(h, r, x, y) = \sum_{j \in \mathcal{J}} \left( c_{\max} - l(y, \widehat{y}_{<j}) - \tilde{c}_j(x, \widehat{y}_{<j}, y) \right) \psi(\boldsymbol{g}(x), j)$$

where $\psi(\boldsymbol{g}(x), j)$ is a consistent surrogate for the multiclass 0-1 loss or $\mathbf{1}_{r(x) \neq j}$, and $c_{\max} = \max_{j \in \mathcal{J}} l(y, \widehat{y}_{\leq j}) + \tilde{c}_j(x, \widehat{y}_{<j}, y)$. Examples of $\psi$ include cross entropy loss, i.e. $\psi_{\text{ce}}(\boldsymbol{g}(x), j) = -\log \frac{e^{g_j(x)}}{\sum_{k \in \mathcal{J}} e^{g_k(x)}}$ or mean absolute error, i.e. $\psi_{\text{mae}}(\boldsymbol{g}(x), j) = 1 - \frac{e^{g_j(x)}}{\sum_{k \in \mathcal{J}} e^{g_k(x)}}$ (Ghosh et al., 2017). The $\mathcal{L}^{\psi}$-risk is characterized by $\mathcal{R}_{\psi}(h, r) = \mathbb{E}_{(x,y) \sim \mathcal{P}_{X,Y}} [\mathcal{L}^{\psi}(h, r, x, y)]$.

Intuitively, for a given token position $j$, $c_{\max} - l(y, \widehat{y}_{\leq j}) - \tilde{c}_j(x, \widehat{y}_{<j}, y)$ is large when the subsequence up till $j$ can be predicted correctly and the cost of deferring on the rest of the sequence is small. When this term is large, we desire $\psi(\boldsymbol{g}(x), j)$ to be small, incentivizing the model to choose $j$ as the deferral point to minimize the surrogate loss. This surrogate loss encourages the rejector to pick a token position which also has minimal costs for deferring on the rest of the sequence - a goal the $\mathcal{L}^{\text{OneTime}}$ loss function shares.

Just like $\mathcal{L}^{\phi}$, $\mathcal{L}^{\psi}$ is also convex and upper bounds $\mathcal{L}^{\text{OneTime}}$ up to some scale $\gamma$ (See Appendix B.2 for proof). Theorem 4.1 shows that $\mathcal{L}^{\psi}$ achieves Bayes consistency.

**Theorem 4.1** (Consistency). *Let $\tilde{c}_j(x, \widehat{y}_{<j}, y) \leq \bar{C} < \infty$, $l(y, \widehat{y}_j^h) \leq \bar{l} < \infty$ for all $j \in \mathcal{J}$, and $\psi$ be cross entropy loss $\psi_{ce}$ or mean absolute error $\psi_{mae}$. If there exists $i, j \in \mathcal{J}$ such that $\left| l(y, \widehat{y}_i^h) + \tilde{c}_i(x, \widehat{y}_{<i}, y) - l(y, \widehat{y}_j^h) - \tilde{c}_j(x, \widehat{y}_{<j}, y) \right| > \Delta > 0$, then, for a fixed $h$, $\mathcal{L}^{\psi}$ is a consistent surrogate for $\mathcal{L}^{OneTime}$.*

The proof of the above theorem can be found in Appendix B.4. If the $l(y, \widehat{y}_{\leq j}) + \tilde{c}_j(x, \widehat{y}_{<j}, y)$ terms are equal for all $j$, the weights would be 0. By assuming a non-zero cost difference for at least one pair, we are ensuring that at least one $j$ has a non-zero weight. Such an assumption can be satisfied if there is a sufficient separation in predictive capabilities between the predictor and the expert.

## 4.3 Generalization Bounds

Under the empirical risk minimization framework, the optimal rejector $\widehat{r}_n \in \arg\min_{r \in \mathcal{F}} \widehat{\mathcal{R}}_{\psi}(r)$ minimizes the empirical risk with respect to $\mathcal{L}^{\psi}$ or $\widehat{\mathcal{R}}_{\psi}(r) = \frac{1}{n} \sum_{(x^i, y^i) \in \mathcal{D}} \mathcal{L}^{\psi}(h, r, x^i, y^i)$ from a finite hypothesis class $\mathcal{F}$. Theorem 4.2 presents generalization bounds for $\mathcal{L}^{\psi}$ (See Appendix B.5 for the proof).

**Theorem 4.2.** *Suppose $r_n \in \mathcal{F} = \{x \to \arg\min_{j \in \mathcal{J}} g_j(x) : g_j \in \mathcal{G}_j\}$ where $\mathcal{G}_j$ consists of functions having a range of $(-M, M)$ with $M \geq 0$, $\tilde{c}_j(x, \widehat{y}_{<j}, y) \leq \bar{C}$, and $l(y, \widehat{y}_j^h) \leq \bar{l} < \infty$. If $\psi$ is $\rho$-lipschitz continuous*

with respect to $\boldsymbol{g}(x)$ and there exists $u_\psi(M)$ such that $\psi\left(\boldsymbol{g}(x),j\right) \leq u_\psi(M) < \infty$ for all $j$ and $r \in \mathcal{F}$, then with probability $1 - \delta$ for a fixed h, the following upper bound holds for the empirical risk minimizer $\widehat{r}_n$ with respect to $\mathcal{L}_\psi^{Point}$:

$$\mathcal{R}_\psi\left(\widehat{r}_n, h\right) - \mathcal{R}_\psi^*\left(\mathcal{F}, h\right) \leq 2\sqrt{2}q(L)\rho \sum_{j \in \mathcal{J}} \mathfrak{R}_n\left(\mathcal{G}_j\right) + u_\psi(M)q(L)\sqrt{\frac{2\log\left(\frac{2}{\delta}\right)}{n}}$$

where $\mathcal{R}_\psi^*\left(\mathcal{F}, h\right) = \inf_{r \in \mathcal{F}} \mathcal{R}_\psi\left(r, h\right)$, $q(L) = 2(|\mathcal{J}| - 1)(\bar{C} + \bar{l})$, and $\mathfrak{R}_n\left(\mathcal{G}_j\right)$ is the Rademacher Complexity of $\mathcal{G}_j$.

Restricting the hypothesis class to have a range $(-M, M)$ practically involves ReLU activation and softmax clipping to prevent the scores $\boldsymbol{g}$ from approaching $-\infty$ or $\infty$. For $\psi_{\mathrm{ce}}$, we can expect $u_{\psi_{\mathrm{ce}}}(M) = -\log\frac{e^{-M}}{e^{-M} + Le^M}$ and for $\psi_{\mathrm{mae}}$, it is $u_{\psi_{\mathrm{mae}}}(M) = 1 - \frac{e^{-M}}{e^{-M} + Le^M}$. We can further express the upper bound of the excess risk respect to the original token loss $\mathcal{L}^{\mathrm{OneTime}}$. The characterization of the upper bound and the proof details can be found in Appendix B.6.

## 5 Related Works

*Model Cascades* (Viola & Jones, 2001) share similarities with the learning to defer framework. In cascades, an inference task is progressively handled by a series of models where easier tasks are completed by earlier models in the sequence and complex cases escalate to subsequent models. Similar to confidence-based L2D methods described in Section 2.1, cascading decisions are typically determined by a thresholding deferral rule (Yue et al., 2023). Some recent work on applying cascades to large language models (Gupta et al., 2024; Wang et al., 2024) has also explored modeling the cascade decision maker. However, model cascades, just like learning to defer methods, ultimately rely on a single model for each prediction. Our work extends cascades by introducing partial deferrals, allowing two models to collaborate on a prediction. Narasimhan et al. (2024) also proposes a token-level deferral rule, but their deferral rule compares the predictor's confidence with expert's confidence at every token position, which requires many expensive expert queries. Our work learns the expert costs at training time to avoid such expert queries unless the rejector calls for it.

*Query Routing* methods select a model that provide the highest quality prediction from an ensemble of models. Some query routing algorithms in large language models (Jiang et al., 2023; Chen et al., 2023) often require querying either a part or the entire model ensemble to determine the best generation. Some methods train query routing models using specially designed loss functions (Shnitzer et al., 2023; Lu et al., 2023; Ding et al., 2024); these loss functions, however, lack consistency guarantees. Our work not only presents surrogate losses with stronger statistical guarantees but also offers more granular routing decisions.

Many L2D methods draw inspiration from the *Cost-Sensitive learning* framework. In the example-dependent cost-sensitive setting, each feature $x$ is paired with a cost vector. A cost-sensitive classifier must then learn to identify the index of the cost vector that minimizes the expected cost. Classifiers (Abe et al., 2004; Lin, 2015) are often trained with a weighted logistic loss or weighted cross entropy loss function, similar to $\mathcal{L}^{\mathrm{Token}}$ and $\mathcal{L}^{\mathrm{OneTime}}$. However, popular methods (Tu & Lin, 2010; Chung et al., 2015) use losses that are inconsistent with the original cost-sensitive loss.

*Selective classification* equips a predictor with an abstention option that trades coverage for accuracy. In the noise-free setting, El-Yaniv & Wiener (2010) formalizes the risk–coverage framework. Deep variants implement the selection function via post-hoc confidence thresholding (Geifman & El-Yaniv, 2017) or joint learning(Geifman & El-Yaniv, 2019). Similarly, *selective regression* extends the reject option to regression (Shah et al., 2022; Jiang et al., 2020; Zaoui et al., 2020). However, unlike learning to defer, these methods treat abstention as a terminal action with a fixed penalty and do not account for the downstream effects of abstention, e.g., accuracy of expert predictions.

## 6 Experiments

Our experiments are designed to evaluate two questions: (i) how partial deferral compares to whole-sequence deferral , and (ii) how our model-based approach compares to confidence-based partial deferral. To this end, we consider three tasks (TSP, XSUM, MWP). We refer the reader to Appendix D for training and hyperparameter details of the model rejectors.

### 6.1 Tasks

**Traveling Salesman Problem (TSP)**: TSP is a classic NP-Hard optimization problems which requires finding the shortest tour visiting all cities exactly once and returning to the start. Optimization-based solvers achieve near-optimality at high computational cost, while ML models are efficient but less accurate. This task highlights a natural collaboration point for partial deferral. We use a PointerNet predictor (Vinyals et al., 2015), trained on synthetic 50-city TSP graphs (coordinates drawn from a standard normal distribution), and a Gurobi solver as the expert. The expert takes the partial solution from the predictor as a constraint and completes the tour. Tours are evaluated by total distance. Since Gurobi does not operate autoregressively, token-level deferral is unsuitable here; instead, this is a prime use case for one-time deferral. The loss $l(y, \widehat{y}_{<j})$ measures the distance of subtour $\widehat{y}_{<j}$, while the expert cost $\tilde{c}_j(x, \widehat{y}_{<j}, y)$ combines the distance of the expert-completed subtour $\widehat{y}^e_{\geq j}$ with a deferral cost $\alpha_j$, which increases monotonically in the number of rejected tokens (see Appendix F).

**Extreme Summarization (XSUM).** The predictor summarizes long articles in a few sentences using a `t5-small` model, and the expert is a stronger `t5-base` model; both are fine-tuned on XSUM (Narayan et al., 2018). We use T5 (encoder–decoder) models because they are more amenable to continue from an arbitrary target prefix $\widehat{y}_{<j}$ while conditioning on the source article through a fixed encoder representation. Although the expert recomputes its own encoder and decoder states after a handoff, the prompt encoding is performed once for the predictor and expert, whereas in decoder-only architectures the prompt forms part of the autoregressive context and typically requires replaying the prompt together with the generated prefix when switching models, resulting in additional computational overhead. Generated summaries are limited to 64 tokens and evaluated with ROUGE-1. For token-level deferral, the predictor loss is $l(y, \widehat{y}^h_j) = \mathbf{1}_{\widehat{y}^h_j \notin y}$, while the expert cost is $c_j(x, \widehat{y}_{<j}, y) = \mathbf{1}_{e(x, \widehat{y}_{<j}) \notin y}$. For one-time deferral, we measure $1 - \text{ROUGE1}$ score of the expert continuation from step $j$, appended to $\widehat{y}_{<j}$. The joint term $l(y, \widehat{y}_{\leq j}) + \tilde{c}_j(x, \widehat{y}_{<j}, y)$ is the sum of this error and the deferral cost $\alpha_j$, as in the TSP case.

**Multi-Step Weather Prediction (MWP)**: Given 2 hours of temperature history (12 observations at 10-minute intervals), the task is to predict the next 2 hours ($L = 12$). The predictor is an LSTM trained on real data from the Max Planck Institute in Jena, Germany. The expert is a simulated human: ground-truth values perturbed with zero-mean Gaussian noise. For token-level deferral, the predictor loss is mean-squared error $l(y, \widehat{y}^h_j) = (\widehat{y}^h_j - y_j)^2$, and the expert cost is $c_j(x, \widehat{y}_{<j}, y) = (e(x, \widehat{y}_{<j}) - y_j)^2$. For one-time deferral, the loss is $l(y, \widehat{y}_{<j}) = \|y_{<j} - \widehat{y}_{<j}\|^2_2$ and $\tilde{c}_j(x, \widehat{y}_{<j}, y)$ combines this squared error of the expert subsequence $\widehat{y}^e_{\geq j}$ and the query cost $\alpha_j$.

### 6.2 Baselines

To compare the cost-accuracy tradeoffs achieved by partial deferral compared with whole-sequence deferral, we consider three confidence-thresholding baselines—`ChowSum`, `ChowMean`, and `ChowQuantile`—which defer the entire sequence based on the sum, mean, or a chosen quantile of token-level confidence scores computed over the predicted sequence. For TSP and XSUM ($|\mathcal{V}| < \infty$), the per-token score is the negative log–softmax; for MWP ($\mathcal{V} = \mathbb{R}$), it is the Monte Carlo dropout variance. `WholeModelScore` and `WholeModelEmbed` are our implementations of the PostHocQuantile and PostHocEmbed methods of Gupta et al. (2024), respectively.

For partial deferral, we compare our model-based rejectors against confidence-based methods at both granularities. At the token level, `TokenwiseScore` defers by thresholding token-level confidence scores, and `TokenwiseEntropy` defers by thresholding the entropy of the predictive distribution. In the one-time setting, `OneTimeScore` and `OneTimeEntropy` select a single handoff position based on the highest token-level

Table 1: Area under the deferral curve (AUDC; lower is better) and percentage improvement relative to a random rejector on TSP, XSUM, and MWP. *Random* flips a biased coin to defer; *Optimal* is an oracle that minimizes loss at each deferral budget. Bold indicates the best result (ties bolded) within each block (Whole, One-time, Token-level). Values are mean (std) over ten runs (200 test instances per run). `OneTimeModel` uses the best-performing $\mathcal{L}^\psi$ variant (see Figure 2).

| | TSP | | XSUM | | MWP | |
|---|---|---|---|---|---|---|
| Method | AUDC | % Improvement | AUDC | % Improvement | AUDC | % Improvement |
| Random | 611.32(16.83) | 0.00 (0.00) | 8940.43(82.11) | 0.00 (0.00) | 12605.29 (3878.81) | 0.00 (0.00) |
| Optimal | 0.00(0.00) | 100.00 (0.00) | 7755.67(109.98) | 13.25(0.66) | 280.48 (20.00) | 97.56 (0.76) |
| | | | Whole Deferrals | | | |
| ChowMean | **577.97(15.51)** | **5.45(1.23)** | 8957.48(73.16) | -0.20(1.53) | **11902.34 (2905.11)** | **-5.60 (56.62)** |
| ChowSum | 577.97(15.51) | 5.45(1.23) | 8927.90(74.97) | 0.13(1.42) | 11902.34 (2905.11) | -5.60 (56.62) |
| ChowQuantile 0 | 610.88(18.39) | 0.07(1.43) | 8963.93(59.99) | -0.28(1.44) | 12216.38 (3226.06) | -7.54 (56.07) |
| ChowQuantile 0.4 | 579.00(15.34) | 5.28(1.03) | 8959.38(71.87) | -0.22(1.44) | 12216.38 (3226.06) | -7.54 (56.07) |
| ChowQuantile 0.8 | 583.87(16.32) | 4.48(1.40) | 8952.24(78.98) | -0.15(1.61) | 12216.38 (3226.06) | -7.54 (56.07) |
| ChowQuantile 1.0 | 597.10(17.41) | 2.32(1.14) | 8935.74(77.52) | 0.04(1.55) | 12216.38 (3226.06) | -7.54 (56.07) |
| WholeModelEmbed | 620.25(22.67) | -1.44(1.21) | 8941.40(51.15) | -0.02(1.37) | 12159.75 (2418.01) | -5.81 (44.85) |
| WholeModelScore | 611.02(19.90) | 0.05(1.41) | **8926.57(51.77)** | **0.15(1.22)** | 13082.56 (3238.24) | -14.38 (53.66) |
| | | | One-time Deferrals | | | |
| OneTimeModel | **562.37(13.36)** | **7.98(1.72)** | **8841.60(41.58)** | **1.10(1.06)** | **8886.51 (1817.31)** | **30.76 (4.86)** |
| OneTimeScore | 644.07(19.64) | -5.37(2.12) | 9305.36(74.03) | -4.09(1.46) | 12681.71 (3650.90) | 3.07 (9.56) |
| OneTimeEntropy | 641.51(19.75) | -4.95(2.26) | 9302.72(75.13) | -4.06(1.47) | - | - |
| | | | Token-level Deferrals | | | |
| TokenwiseModel | - | - | **8073.89(96.57)** | **9.69(0.50)** | 10230.69 (3764.91) | 19.78 (6.61) |
| TokenwiseScore | - | - | 8166.72(102.27) | 8.66(0.60) | 14688.00 (3994.59) | -18.06 (9.68) |
| TokenwiseEntropy | - | - | 8200.95(99.10) | 8.27(0.54) | - | - |

confidence and predictive entropy, respectively. Entropy-based baselines apply only to XSUM and TSP, where the output vocabulary $\mathcal{V}$ is finite. Formal expressions of these baselines are provided in Appendix C.

## 6.3 Evaluation

We evaluate our methods using *deferral curves*, which plot the system loss against the number of deferred tokens using various thresholds of the rejector score $r(x)$(see Figure 1). Loss is task-specific: for TSP, it is the percentage increase in tour length relative to the expert; for XSUM, it is $1 - \text{ROUGE1}$; and for MWP, it is the squared error $\|y - \hat{y}\|_2^2$. The area under the deferral curve (AUDC) summarizes the cost-quality tradeoff, and we also report the percentage improvement in AUDC over a random rejector, whose trade-off is linear. The resulting deferral curves should therefore be interpreted under the task-specific cost models defined for each experiment. They do not measure end-to-end deployment quantities such as inference latency, API cost, or monetary cost, which depend on the underlying deployment architecture. Thresholds do not directly determine rejection decisions in the one-time setting; to obtain similar curves, we adopt the following policy: if $g_{L+1}(x)$ (the score of the "no deferral" action) exceeds a threshold, the predictor completes the sequence, otherwise we defer at $\arg\max_{j \in \mathcal{J}} g_j(x)$.

All results are averaged over ten runs, each with 200 test instances. For XSUM, we evaluate on a random subset (2,000 instances) of the test set to keep repeated evaluation across ten runs and multiple rejector baselines computationally tractable. Although evaluation on the full test set would provide a more comprehensive assessment, the low variability across runs suggests that the qualitative conclusions are unlikely to change.

## 6.4 Comparison with other deferral methods

Table 1 shows that model-based token-level and one-time deferral schemes outperform whole-sequence baselines (both model- and confidence-based); where applicable, token-level deferral attains the best AUDC. Within each type of granularity, model-based rejectors consistently exceed their confidence-thresholding counterparts. In XSUM, even confidence-based token-level deferral surpasses all whole-sequence methods,

highlighting the advantage of fine-grained deferral. Gains for one-time deferral on XSUM are modest . By contrast, in TSP and MWP, one-time deferral exhibits cost–accuracy trade-offs.

The behavior of one-time deferral depends on the assumed per-token expert cost $\alpha_j$. Larger $\alpha_j$ penalize earlier handoffs more heavily, whereas smaller per-token costs make earlier expert intervention more favorable. We study the effects of $\alpha_j$ on the balance between prediction quality and expert usage in Appendix F.

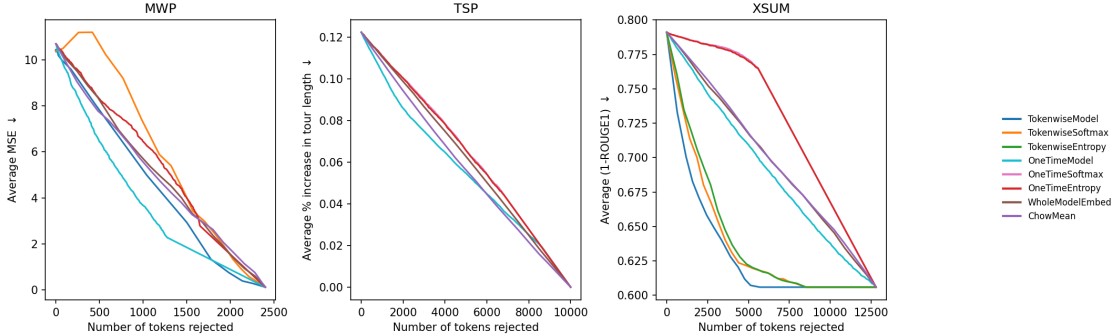

Figure 1: Deferral Curve for MWP, TSP, and XSUM (closer to the lower left corner is better).

### 6.5 Size of $\mathcal{J}$ for One time Deferral

While allowing every token position as a candidate handoff position provided maximum flexibility, it also increases the complexity of the multiclass problem since the rejector must distinguish between many deferral points that induce similar predictor-expert collaborations. To highlight this, we train one-time rejectors with $\mathcal{L}^\psi$ with varying candidate sets of deferral points $\mathcal{J}$ to assess how the number of allowable handoff positions affects cost–effectiveness. For these tasks, $\mathcal{J}$ is a uniform grid over $\{1, \ldots, L+1\}$ (with $L+1$ denoting "no deferral"), and we vary the grid granularity. Figure 2 demonstrates the need for a modified one-time deferral loss function. The model trained with all token positions ($|\mathcal{J}| = L + 1$) is not always the most optimal choice, with this effect exacerbated in the XSUM and TSP experiments. One-time rejectors are generally more cost-effective than its corresponding confidence-based rejectors regardless of the size of $\mathcal{J}$.

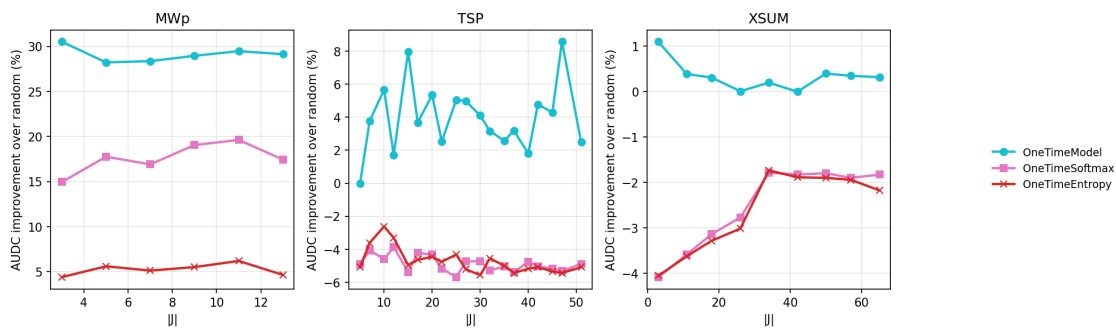

Figure 2: Plots of Size of $\mathcal{J}$ against percentage improvement in AUDC relative to the random baseline for MWP (on the left) ,TSP (middle), XSUM (on the right). See Appendix G for exact AUDC values.

## 7  Discussion

We have introduced partial rejections as a way to improve the cost-effectiveness of a learning to defer system and proposed two model-based methods to support granular rejections. This work raises numerous interesting questions. We have currently proposed partial deferral with access to one expert, however, it would

be interesting to extend this to the case with multiple experts. While our language-model experiments use encoder-decoder architectures, extending partial deferral to modern decoder-only LLMs requires accounting for architecture-specific inference costs, including autoregressive replay and model handoff overhead. Accordingly, our empirical results should be interpreted as validating the proposed partial-deferral framework under the encoder-decoder setting rather than demonstrating deployment savings for decoder-only systems. Another important direction is reducing the computational overhead of model handoffs~~, particularly for decoder-only models~~. Our current implementation requires the model to reconstruct its autoregressive state after each handoff. Future work could instead investigate architectures that share intermediate embeddings like key value caches and hidden states between predictor and expert models, enabling more efficient partial feedback. It would also be valuable to explore other types of partial deferrals with granularities between token-level and one-time deferrals. A rejector, for example, could select contiguous parts of a sequence to reject. Another interesting direction is reducing the adaptation cost of the rejector across different predictor–expert pairs. Our current rejectors are trained for a fixed predictor, expert, and task, requiring retraining whenever one of these components changes. Future work could investigate meta-learning or transferable rejectors that rapidly adapt across predictor–expert pairs while reducing the need for pair-specific retraining. Finally, studying partial rejections for larger outputs like protein structures would also be insightful.

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

# A    Proofs for Section 3

## A.1    Proof of Proposition A.1

**Proposition A.1.** *If $\phi$ is binary classification calibrated and convex, $\mathcal{L}^{\phi}(h, r, x, y)$ is a convex upper bound of $\mathcal{L}^{Token}(h, r, x, y)$ upto some scale $\gamma$*

*Proof.* It is easy to see that $\mathcal{L}_j^{\phi}(h, r, x, y)$ is convex with respect to $r$ as long as $\phi(\cdot)$ is convex. Since $\phi$ is binary classification calibrated, there is a constant $\gamma$ such that $\gamma\phi(z) \geq \mathbf{1}_{z \leq 0}$ by Lemma 3 from Bartlett et al. (2006). Therefore, $\gamma\mathcal{L}^{\phi}(h, r, x, y) \geq \mathcal{L}^{\text{Token}}(h, r, x, y)$.

$\square$

## A.2    Proof of Lemma 3.1

**Lemma A.2.** *Let $0 < \bar{c} \leq c_j(x, \widehat{y}_{<j}, y) \leq \bar{C} < \infty$ for all $j$, $0 \leq l(y, \widehat{y}_j^h) \leq \bar{l} < \infty$, and $\phi$ be a binary surrogate loss that satisfies the following inequality for any distribution over $\mathcal{X} \times \{-1, 1\}$ and any measurable function $f$:*

$$\mathcal{R}_{binary\ 0-1}(f) - \mathcal{R}^*_{binary\ 0-1} \leq \Gamma\left(\mathcal{R}_{binary\ \phi}(f) - \mathcal{R}^*_{binary\ \phi}\right)$$

*where $\mathcal{R}_{binary\ 0-1}$ is the binary 0-1 risk and $\Gamma : \mathbb{R}^+ \to \mathbb{R}^+$ is a non-decreasing concave function. Then, for any measurable $r$ and any $h$, the following inequality holds for any distribution over $\mathcal{X} \times \mathcal{Y}$:*

$$\mathcal{R}_{Token}(r, h) - \mathcal{R}^*_{Token}(h) \leq \tilde{\Gamma}\left(\mathcal{R}_{\phi}(r, h) - \mathcal{R}^*_{\phi}(h)\right)$$

*where $\tilde{\Gamma}(z) = \left(\bar{l} + \bar{C}\right)\Gamma\left(\frac{z}{\bar{c}}\right)$, $\mathcal{R}^*_{Token}(h) := \inf_r \mathcal{R}_{Token}(r, h)$, and $\mathcal{R}^*_{\phi}(h) := \inf_r \mathcal{R}_{\phi}(r, h)$*

*Proof.* Let $C_{\text{Token}}(r, h, x) = \mathbb{E}_{y \sim \mathcal{P}_{Y|X=x}}\left[\mathcal{L}^{\text{Token}}(h, r, x, y)\right]$ be the pointwise $\mathcal{L}^{\text{Token}}$- risk for a given $x$ and predictor $h$ and let $C^*_{\text{Token}}(h, x) = \inf_r C_{\text{Token}}(r, h, x)$ be the optimal pointwise $\mathcal{L}^{\text{Token}}$- risk for a given $x$ and predictor $h$. Similarly, let $C_{\phi}(r, h, x) = \mathbb{E}_{y \sim \mathcal{P}_{Y|X=x}}\left[\mathcal{L}^{\phi}(h, r, x, y)\right]$ be the pointwise $\mathcal{L}^{\phi}$- risk for a given $x$ and predictor $h$ and $C^*_{\phi}(h, x) = \inf_r C_{\phi}(r, h, x)$ be the optimal pointwise $\mathcal{L}^{\phi}$- risk. For any measurable function $r$, the excess pointwise $\mathcal{L}^{\phi}$- risk can be characterized by:

$$
\begin{aligned}
&C_{\text{Token}}(r, h, x) - C^*_{\text{Token}}(h, x) \\
&= \frac{1}{L}\sum_{j=1}^{L} \mathbb{E}_{y \sim \mathcal{P}_{Y|X=x}}\left[l\left(y, \widehat{y}_j^h\right)\right]\mathbf{1}_{r(x, \widehat{y}_{<j}) \leq 0} + \mathbb{E}_{y \sim \mathcal{P}_{Y|X=x}}\left[c_j(x, \widehat{y}_{<j}, y)\right]\mathbf{1}_{r(x, \widehat{y}_{<j}) > 0} \\
&\quad - \frac{1}{L}\inf_{\tilde{r}}\left\{\sum_{j=1}^{L} \mathbb{E}_{y \sim \mathcal{P}_{Y|X=x}}\left[l\left(y, \widehat{y}_j^h\right)\right]\mathbf{1}_{\tilde{r}(x, \widehat{y}_{<j}) \leq 0} + \mathbb{E}_{y \sim \mathcal{P}_{Y|X=x}}\left[c_j(x, \widehat{y}_{<j}, y)\right]\mathbf{1}_{\tilde{r}(x, \widehat{y}_{<j}) > 0}\right\}
\end{aligned}
$$

Let $A_j = \mathbb{E}_{y \sim \mathcal{P}_{Y|X=x}} [l(y, \widehat{y}_{<j})]$ and $B_j = \mathbb{E}_{y \sim \mathcal{P}_{Y|X=x}} [c_j(x, \widehat{y}_{<j}, y)]$

$$C_{\text{Token}}(r, h, x) - C^*_{\text{Token}}(h, x)$$

$$= \frac{1}{L} \sum_{j=1}^{L} A_j \mathbf{1}_{r_j(x, \widehat{y}_{<j}) \le 0} + B_j \mathbf{1}_{r_j(x, \widehat{y}_{<j}) > 0} - \frac{1}{L} \inf_{\tilde{r}} \left\{ \sum_{j=1}^{L} A_j \mathbf{1}_{\tilde{r}_j(x, \widehat{y}_{<j}) \le 0} + B_j \mathbf{1}_{\tilde{r}_j(x, \widehat{y}_{<j}) > 0} \right\}$$

$$= \frac{1}{L} \sum_{j=1}^{L} (A_j + B_j) \left( \frac{A_j}{A_j + B_j} \mathbf{1}_{r_j(x, \widehat{y}_{<j}) \le 0} + \frac{B_j}{A_j + B_j} \mathbf{1}_{r_j(x, \widehat{y}_{<j}) > 0} \right)$$

$$- \frac{1}{L} \inf_{\tilde{r}} \left\{ \sum_{j=1}^{L} (A_j + B_j) \left( \frac{A_j}{A_j + B_j} \mathbf{1}_{\tilde{r}_j(x, \widehat{y}_{<j}) \le 0} + \frac{B_j}{A_j + B_j} \mathbf{1}_{\tilde{r}_j(x, \widehat{y}_{<j}) > 0} \right) \right\}$$

$$\le \frac{(\bar{l} + \bar{C})}{L} \sum_{j=1}^{L} \frac{A_j}{A_j + B_j} \mathbf{1}_{r_j(x, \widehat{y}_{<j}) \le 0} + \frac{B_j}{A_j + B_j} \mathbf{1}_{r_j(x, \widehat{y}_{<j}) > 0}$$

$$- \frac{(\bar{l} + \bar{C})}{L} \inf_{\tilde{r}} \left\{ \sum_{j=1}^{L} \frac{A_j}{A_j + B_j} \mathbf{1}_{\tilde{r}_j(x, \widehat{y}_{<j}) \le 0} + \frac{B_j}{A_j + B_j} \mathbf{1}_{\tilde{r}_j(x, \widehat{y}_{<j}) > 0} \right\}$$

Consider a feature space $\tilde{\mathcal{X}} = \mathcal{X} \times \mathcal{V}^*$ and output space $\tilde{\mathcal{Y}} = \{-1, 1\}$. Let the distribution on $\tilde{X}$ be uniform over $\{\tilde{x}_1, \cdots, \tilde{x}_L\}$ where $\tilde{x}_j = (x, \widehat{y}_{<j})$ with $\frac{1}{L}$ mass on each point on the set. Let $P(\tilde{Y} = -1 \mid \tilde{X} = \tilde{x}_j) = \frac{A_j}{A_j + B_j}$ and $f$ be a measurable function such that $f(\tilde{x}_j) = r_j(\tilde{x}_j) = r_j(x, \widehat{y}_{<j})$

Under this distribution,

$$\mathcal{R}_{\text{binary } 0-1}(r) = \frac{1}{L} \sum_{j=1}^{L} \frac{A_j}{A_j + B_j} \mathbf{1}_{r_j(x, \widehat{y}_{<j}) \le 0} + \frac{B_j}{A_j + B_j} \mathbf{1}_{r_j(x, \widehat{y}_{<j}) > 0}$$

$$\mathcal{R}^*_{\text{binary } 0-1} = \frac{1}{L} \inf_{\tilde{r}} \left\{ \sum_{j=1}^{L} \frac{A_j}{A_j + B_j} \mathbf{1}_{\tilde{r}_j(x, \widehat{y}_{<j}) \le 0} + \frac{B_j}{A_j + B_j} \mathbf{1}_{\tilde{r}_j(x, \widehat{y}_{<j}) > 0} \right\}$$

$$\mathcal{R}_{\text{binary } \phi}(r) = \frac{1}{L} \sum_{j=1}^{L} \frac{A_j}{A_j + B_j} \phi(r_j(x, \widehat{y}_{<j})) + \frac{B_j}{A_j + B_j} \phi(-r_j(x, \widehat{y}_{<j}))$$

$$\mathcal{R}^*_{\text{binary } \phi} = \frac{1}{L} \inf_{\tilde{r}} \left\{ \sum_{j=1}^{L} \frac{A_j}{A_j + B_j} \phi(\tilde{r}_j(x, \widehat{y}_{<j})) + \frac{B_j}{A_j + B_j} \phi(-\tilde{r}_j(x, \widehat{y}_{<j})) \right\}$$

From the theorem statement,

$$C_{\text{Token}}(r, h, x) - C^*_{\text{Token}}(h, x)$$

$$\leq \frac{(\bar{l} + \bar{C})}{L} \sum_{j=1}^{L} \frac{A_j}{A_j + B_j} \mathbf{1}_{r_j(x, \widehat{y}_{<j}) \leq 0} + \frac{B_j}{A_j + B_j} \mathbf{1}_{r_j(x, \widehat{y}_{<j}) > 0}$$

$$- \frac{(\bar{l} + \bar{C})}{L} \inf_{\tilde{r}} \left\{ \sum_{j=1}^{L} \frac{A_j}{A_j + B_j} \mathbf{1}_{\tilde{r}_j(x, \widehat{y}_{<j}) \leq 0} + \frac{B_j}{A_j + B_j} \mathbf{1}_{\tilde{r}_j(x, \widehat{y}_{<j}) > 0} \right\}$$

$$\leq (\bar{l} + \bar{C}) \Gamma \left( \sum_{j=1}^{L} \frac{A_j \phi(r_j(x, \widehat{y}_{<j})) + B_j \phi(-r_j(x, \widehat{y}_{<j}))}{L(A_j + B_j)} - \inf_{\tilde{r}} \left\{ \sum_{j=1}^{L} \frac{A_j \phi(\tilde{r}_j(x, \widehat{y}_{<j})) + B_j \phi(-\tilde{r}_j(x, \widehat{y}_{<j}))}{L(A_j + B_j)} \right\} \right)$$

$$\leq (\bar{l} + \bar{C}) \Gamma \left( \sum_{j=1}^{L} \frac{A_j \phi(r_j(x, \widehat{y}_{<j})) + B_j \phi(-r_j(x, \widehat{y}_{<j}))}{L\bar{c}} - \inf_{\tilde{r}} \left\{ \sum_{j=1}^{L} \frac{A_j \phi(\tilde{r}_j(x, \widehat{y}_{<j})) + B_j \phi(-\tilde{r}_j(x, \widehat{y}_{<j}))}{L\bar{c}} \right\} \right)$$

$$= (\bar{l} + \bar{C}) \Gamma \left( \frac{C_\phi(r, h, x) - C^*_\phi(h, x)}{\bar{c}} \right)$$

Let $\tilde{\Gamma}(z) = (\bar{l} + \bar{C}) \Gamma \left( \frac{z}{\bar{c}} \right)$. Then,

$$\begin{aligned}
\mathcal{R}_{\text{Token}}(r, h) - \mathcal{R}^*_{\text{Token}}(h) &= \mathbb{E}_{x \sim \mathcal{P}_X} \left[ C_{\text{Token}}(r, h, x) - C^*_{\text{Token}}(h, x) \right] \\
&\leq \mathbb{E}_{x \sim \mathcal{P}_X} \left[ \tilde{\Gamma} \left( C_\phi(r, h, x) - C^*_\phi(h, x) \right) \right] \\
&\overset{(a)}{\leq} \tilde{\Gamma} \left( \mathbb{E}_{x \sim \mathcal{P}_X} \left[ C_\phi(r, h, x) - C^*_\phi(h, x) \right] \right) \\
&\overset{(b)}{=} \tilde{\Gamma} \left( \mathcal{R}_\phi(r.h) - \mathcal{R}^*_\phi(h) \right)
\end{aligned}$$

(a) by Jensen's inequality since $\tilde{\Gamma}$ is concave, (b) since the infimum is taken over all measurable functions $\mathcal{R}^*_\phi(h) = \mathbb{E}_{x \sim \mathcal{P}_X} \left[ C^*_\phi(h, x) \right]$

$\square$

### A.3 Proof of Theorem 3.2

**Definition A.3** (Classification Calibration). Let $\eta(x) = \mathbb{P}[Y = 1 \mid X = x]$ and $\mathcal{R}_\phi(f) = \mathbb{E}_{x \sim \mathcal{P}_X} [C_\phi(\eta(x), f(x))]$ where $C_\phi(\eta, t) = \eta \phi(t) + (1 - \eta) \phi(-t)$. $\phi$ is considered binary classification calibrated if $C^-_\phi(\eta) - C^*_\phi(\eta) > 0$ for all $\eta \neq \frac{1}{2}$ where $C^-_\phi(\eta) = \inf_{t:t(\eta - \frac{1}{2}) \leq 0} C_\phi(\eta, t)$ and $C^-_\phi(\eta) = \inf_{t \in \mathbb{R}} C_\phi(\eta, t)$

**Theorem A.4** (Consistency). *If $0 < \bar{c} \leq c_j(x, \widehat{y}_{<j}, y) \leq \bar{C} < \infty$ for all $j$, $0 \leq l(y, \widehat{y}^h_j) \leq \bar{l} < \infty$, and $\phi$ be binary classification calibrated, then, for a fixed $h$, $\mathcal{L}^\phi$ is a Bayes consistent surrogate for $\mathcal{L}^{Token}$*

*Proof.* Since $\phi$ is binary classification calibrated, we know that $\mathcal{R}_{\text{binary } 0-1}(f) - \mathcal{R}^*_{\text{binary } 0-1} \leq \psi^{-1} \left( \mathcal{R}_{\text{binary } \phi}(f) - \mathcal{R}^*_{\text{binary } \phi} \right)$ where $\psi$ is a non-decreasing convex function, making $\psi^{-1}$ a non-decreasing concave function Bartlett et al. (2006). By Lemma 3.1, $\mathcal{R}_{\text{Token}}(r, h) - \mathcal{R}^*_{\text{Token}}(h) \leq \tilde{\Gamma} \left( \mathcal{R}_\phi(r, h) - \mathcal{R}^*_\phi(h) \right)$

where $\tilde{\Gamma}(z) = \left(1 + \bar{C}\right)\psi^{-1}\left(\frac{z}{\bar{c}}\right)$. If $\lim_{n \to \infty} \mathcal{R}_\phi\left(r_n, h\right) - \mathcal{R}_\phi^*(h) = 0$, then

$$
\begin{aligned}
\lim_{n \to \infty} \mathcal{R}_{\text{Token}}\left(r_n, h\right) - \mathcal{R}_{\text{Token}}^*(h) &\leq \lim_{n \to \infty} \tilde{\psi}\left(\mathcal{R}_\phi\left(r_n, h\right) - \mathcal{R}_\phi^*(h)\right) \\
&\overset{(a)}{=} \left(1 + \bar{C}\right)\psi\left(\lim_{n \to \infty} \frac{\mathcal{R}_\phi\left(r_n, h\right) - \mathcal{R}_\phi^*(h)}{\bar{c}}\right) \\
&= \left(1 + \bar{C}\right)\psi(0) \\
&\overset{(b)}{=} 0
\end{aligned}
$$

(a) by continuity of $\psi$ at 0 and (b) since $\psi(0) = 0$ Bartlett et al. (2006) $\qquad \square$

## A.4 Proof of Theorem 3.3

**Theorem A.5.** *Suppose $r \in \mathcal{F}$ which is composed of $\{\mathcal{F}_j\}_{j=1}^L$ such that $r_j \in \mathcal{F}_j$, $c_j(x) \leq \bar{C}$, and $l(y, \widehat{y}_j^h) \leq \bar{l} < \infty$. If $\phi$ is $\rho$-lipschitz continuous, then with probability $1 - \delta$, the following upper bound holds for the empirical risk minimizer $\widehat{r}_n$ with respect to $\mathcal{L}_\phi$ for a fixed $h$:*

$$
\mathcal{R}_\phi\left(\widehat{r}_n, h\right) - \mathcal{R}_\phi^*(\mathcal{F}, h) \leq \frac{4\rho\sqrt{2\left(\bar{C}^2 + \bar{l}^2\right)}}{L}\sum_{j=1}^L \mathfrak{R}_n\left(\mathcal{F}_j\right) + \left(\bar{C} + \bar{l}\right)\sqrt{\frac{2\log\left(\frac{2}{\delta}\right)}{n}}
$$

*where $\mathcal{R}_\phi^*(\mathcal{F}, h) = \inf_{r \in \mathcal{F}} \mathcal{R}_\phi\left(r, h\right)$ and $\mathfrak{R}_n\left(\mathcal{F}_j\right)$ is the Rademacher Complexity of $\mathcal{F}_j$*

*Proof.* Let $G_n = \sup_{r \in \mathcal{F}}\left[\mathcal{R}_\phi\left(r, h\right) - \widehat{\mathcal{R}}_\phi(r, h)\right]$, $G_n' = \sup_{r \in \mathcal{F}}\left[\widehat{\mathcal{R}}_\phi(r, h) - \mathcal{R}_\phi\left(r, h\right)\right]$, and $\mathcal{B} = \{(x, y) \mapsto \mathcal{L}^\phi\left(h, r, x, y\right) : r \in \mathcal{F}\}$.

If $z^i = (x^i, y^i)$, $G_n = g(z^1, \cdots, z^n)$. Since $\left|g(z^1, \cdots, z^i, \cdots, z^n) - g(z^1, \cdots, z^{i'}, \cdots, z^n)\right| \leq \frac{\bar{C} + \bar{l}}{n}$, by applying McDiarmid's inequality

$$
\begin{aligned}
P\left(G_n > \frac{\epsilon}{2}\right) &\leq \exp\left\{\frac{-2\left(\frac{\epsilon}{2} - \mathbb{E}\left[G_n\right]\right)^2}{\sum_{i=1}^n \frac{(\bar{C} + \bar{l})^2}{n^2}}\right\} \\
&\leq \exp\left\{\frac{-2n\left(\frac{\epsilon}{2} - 2\mathfrak{R}_n\left(\mathcal{B}\right)\right)^2}{(\bar{C} + \bar{l})^2}\right\} \\
&\overset{\text{def}}{=} \frac{\delta}{2}
\end{aligned}
$$

Similarly,

$$
P\left(G_n' > \frac{\epsilon}{2}\right) \leq \exp\left\{\frac{-2n\left(\frac{\epsilon}{2} - 2\mathfrak{R}_n\left(\mathcal{B}\right)\right)^2}{(\bar{C} + \bar{l})^2}\right\} = \frac{\delta}{2}
$$

Let $\epsilon = 4\mathfrak{R}_n\left(\mathcal{B}\right) + (\bar{C} + \bar{l})\sqrt{\frac{2\log\left(\frac{2}{\delta}\right)}{n}}$

$$
\begin{aligned}
P\left(\mathcal{R}_\phi\left(\widehat{r}_n, h\right) - \mathcal{R}_\phi^*(\mathcal{F}, h) > \epsilon\right) &\leq P\left(\sup_{r \in \mathcal{F}}\left|\mathcal{R}_\phi\left(r, h\right) - \widehat{\mathcal{R}}_\phi(r, h)\right| > \frac{\epsilon}{2}\right) \\
&\leq P\left(G_n > \frac{\epsilon}{2}\right) + P\left(G_n' > \frac{\epsilon}{2}\right) \\
&\leq \delta
\end{aligned}
$$

With at least probability $1 - \delta$, $\mathcal{R}_\phi\left(\widehat{r}_n, h\right) - \mathcal{R}_\phi^*\left(\mathcal{F}, h\right) \leq 4\mathfrak{R}_n\left(\mathcal{B}\right) + \left(\bar{C} + \bar{l}\right)\sqrt{\frac{2\log\left(\frac{2}{\delta}\right)}{n}}$.

We can further bound $\mathfrak{R}_n\left(\mathcal{B}\right)$ by $\mathfrak{R}_n\left(\mathcal{F}_j\right)$ if we show that $\mathcal{L}^\phi$ is Lipschitz continuous with respect to $r$

$$
\left\|\mathcal{L}^\phi(h, r_1, x, y) - \mathcal{L}^\phi(h, r_2, x, y)\right\|_2^2
$$
$$
\leq \left\|\frac{1}{L}\sum_{j=1}^{L} l\left(y, \widehat{y}_j^h\right)\phi(r_{1,j}(x, \widehat{y}_{<j})) - l\left(y, \widehat{y}_j^h\right)\phi(r_{2,j}(x, \widehat{y}_{<j}))\right\|_2^2
$$
$$
+ \left\|\frac{1}{L}\sum_{j=1}^{L} c_j(x, \widehat{y}_{<j})\phi(-r_{1,j}(x, \widehat{y}_{<j})) - c_j(x, \widehat{y}_{<j})\phi(-r_{2,j}(x, \widehat{y}_{<j}))\right\|_2^2
$$
$$
\leq \frac{\bar{l}^2}{L^2}\sum_{j=1}^{L}\left\|\phi(r_{1,j}(x, \widehat{y}_{<j})) - \phi(r_{2,j}(x, \widehat{y}_{<j}))\right\|_2^2 + \frac{\bar{C}^2}{L^2}\sum_{j=1}^{L}\left\|\phi(-r_{1,j}(x, \widehat{y}_{<j})) - \phi(-r_{2,j}(x, \widehat{y}_{<j}))\right\|_2^2
$$
$$
\leq \frac{\rho^2(\bar{C}^2 + \bar{l}^2)}{L^2}\sum_{j=1}^{L}\left\|r_{1,j}(x, \widehat{y}_{<j}) - r_{2,j}(x, \widehat{y}_{<j})\right\|_2^2
$$

By the extension of the Talagrand's Contraction Lemma Maurer (2016),

$$
\mathfrak{R}_n\left(\mathcal{B}\right) \leq \frac{\rho\sqrt{2\left(\bar{C}^2 + \bar{l}^2\right)}}{L}\sum_{j=1}^{L}\mathfrak{R}_n\left(\mathcal{F}_j\right)
$$

$\square$

## A.5  Proof of Corollary 3.4

**Corollary A.6.** *Suppose* $r \in \mathcal{F}$ *which is composed of* $\{\mathcal{F}_j\}_{j=1}^{L}$ *such that* $r_j \in \mathcal{F}_j$, $0 < \bar{c} \leq c_j(x) \leq \bar{C}$, *and* $l(y, \widehat{y}_j^h) \leq \bar{l} < \infty$. *If* $\phi$ *is a* $\rho$-*lipschitz continuous binary classification-calibrated function, then with probability* $1 - \delta$, *the following upper bound on the excess risk with respect to* $\mathcal{L}^{Token}$ *holds for the empirical risk minimizer* $\widehat{r}_n$ *with respect to* $\mathcal{L}_\phi$ *for a fixed* $h$:

$$
\mathcal{R}_{Token}\left(\widehat{r}_n, h\right) - \mathcal{R}_{Token}^*(h) \leq \tilde{\Gamma}\left(\mathcal{B}(\mathcal{F}) + \mathcal{A}_\phi\left(\mathcal{F}, h\right)\right)
$$

*where* $\tilde{\Gamma} : \mathbb{R}^+ \to \mathbb{R}^+$, $\mathcal{B}(\mathcal{F}) = \frac{4\rho\sqrt{2(\bar{C}^2 + \bar{l}^2)}}{L}\sum_{j=1}^{L}\mathfrak{R}_n\left(\mathcal{F}_j\right) + \left(\bar{C} + \bar{l}\right)\sqrt{\frac{2\log\left(\frac{2}{\delta}\right)}{n}}$, *and* $\mathcal{A}_\phi\left(\mathcal{F}, h\right) = \mathcal{R}_\phi^*\left(\mathcal{F}, h\right) - \mathcal{R}_\phi^*(h)$ *is the approximation error.*

*Proof.* Excess risk can be decomposed into estimation error and approximation error

$$
\mathcal{R}_\phi\left(r, h\right) - \mathcal{R}_\phi^*(h) = \underbrace{\mathcal{R}_\phi\left(r, h\right) - \mathcal{R}_\phi^*\left(\mathcal{F}, h\right)}_{\text{Estimation Error}} + \underbrace{\mathcal{R}_\phi^*\left(\mathcal{F}, h\right) - \mathcal{R}_\phi^*(h)}_{\text{Approximation Error}}
$$

From Theorem 3.3, we have an upper bound on the estimation error and so using that, we have the following inequality

$$
\mathcal{R}_\phi\left(r, h\right) - \mathcal{R}_\phi^*(h) \leq \frac{4\rho\sqrt{2\left(\bar{C}^2 + \bar{l}^2\right)}}{L}\sum_{j=1}^{L}\mathfrak{R}_n\left(\mathcal{F}_j\right) + \left(\bar{C} + \bar{l}\right)\sqrt{\frac{2\log\left(\frac{2}{\delta}\right)}{n}} + \mathcal{A}_\phi\left(\mathcal{F}, h\right)
$$

Since $\phi$ is binary classification calibrated, we know that

$$
\mathcal{R}_{\text{binary } 0-1}(f) - \mathcal{R}_{\text{binary } 0-1}^* \leq \psi^{-1}\left(\mathcal{R}_{\text{binary } \phi}(f) - \mathcal{R}_{\text{binary } \phi}^*\right)
$$

where $\psi$ is a non-decreasing convex function, making $\psi^{-1}$ a non-decreasing concave function. The proof is complete by using Lemma 3.1 $\square$

# B  Proofs for Section 4

## B.1  Proof of Lemma B.1

**Lemma B.1.** *For any $h$, $r$, $x \in \mathcal{X}$, $y \in \mathcal{Y}$, the following equality holds*

$$\mathcal{L}^{OneTime}(h,r,x,y) = \sum_{j \in \mathcal{J}} \left(c_{max} - l(y,\widehat{y}_{\leq j}) - \tilde{c}_j(x,\widehat{y}_{<j},y)\right)\mathbf{1}_{r(x) \neq j} - (|\mathcal{J}|-1)c_{max} + \sum_{j \in \mathcal{J}} l(y,\widehat{y}_{\leq j}) + \sum_{j \in \mathcal{J}} \tilde{c}_j(x,\widehat{y}_{<j},y)$$

*Proof.*

$$\mathcal{L}^{\mathrm{OneTime}}(h,r,x,y)$$
$$= \sum_{j \in \mathcal{J}} \left(l(y,\widehat{y}_{\leq j}) + \tilde{c}_j(x,\widehat{y}_{<j},y)\right)\mathbf{1}_{r(x) = j}$$
$$= \sum_{j \in \mathcal{J}} \left(l(y,\widehat{y}_{\leq j}) + \tilde{c}_j(x,\widehat{y}_{<j},y)\right) - \sum_{j \in \mathcal{J}} \left(l(y,\widehat{y}_{\leq j}) + \tilde{c}_j(x,\widehat{y}_{<j},y)\right)\mathbf{1}_{r(x) \neq j}$$
$$= \sum_{j \in \mathcal{J}} \left(l(y,\widehat{y}_{\leq j}) + \tilde{c}_j(x,\widehat{y}_{<j},y)\right) - \sum_{j \in \mathcal{J}} \left(l(y,\widehat{y}_{\leq j}) + \tilde{c}_j(x,\widehat{y}_{<j},y)\right)\mathbf{1}_{r(x) \neq j} + (|\mathcal{J}|-1)c_{\max} - (|\mathcal{J}|-1)c_{\max}$$
$$= \sum_{j \in \mathcal{J}} \left(l(y,\widehat{y}_{\leq j}) + \tilde{c}_j(x,\widehat{y}_{<j},y)\right) - \sum_{j \in \mathcal{J}} \left(l(y,\widehat{y}_{\leq j}) + \tilde{c}_j(x,\widehat{y}_{<j},y)\right)\mathbf{1}_{r(x) \neq j} + \sum_{j \in \mathcal{J}} c_{\max}\mathbf{1}_{r(x) \neq j} - (|\mathcal{J}|-1)c_{\max}$$
$$= \sum_{j \in \mathcal{J}} \left(c_{\max} - l(y,\widehat{y}_{\leq j}) - \tilde{c}_j(x,\widehat{y}_{<j},y)\right)\mathbf{1}_{r(x) \neq j} - (|\mathcal{J}|-1)c_{\max} + \sum_{j \in \mathcal{J}} l(y,\widehat{y}_{\leq j}) + \sum_{j \in \mathcal{J}} \tilde{c}_j(x,\widehat{y}_{<j},y)$$

$\square$

## B.2  Proof of Proposition B.2

**Proposition B.2.** $\mathcal{L}^{\psi}(h,r,x,y)$ *is a convex upper bound of* $\mathcal{L}^{OneTime}(h,r,x,y)$ *for* $\psi \equiv \psi_{ce}$ *and* $\psi \equiv \psi_{mae}$ *upto some scale* $\gamma$ *if* $c_{\max} > \frac{1}{|\mathcal{J}|-1} \sum_{j \in \mathcal{J}} \left(l(y,\widehat{y}_{\leq j}) + \tilde{c}_j(x,\widehat{y}_{<j},y)\right)$

*Proof.* It is easy to see that $\mathcal{L}^{\psi}(h,r,x,y)$ is convex with respect to $r$ as $\psi_{\mathrm{ce}}(\cdot)$ and $\psi_{\mathrm{mae}}(\cdot)$ are convex.

From Lemma B.1, we know that

$$\mathcal{L}^{\mathrm{OneTime}}(h,r,x,y) = \sum_{j \in \mathcal{J}} \left(c_{\max} - l(y,\widehat{y}_{\leq j}) - \tilde{c}_j(x,\widehat{y}_{<j},y)\right)\mathbf{1}_{r(x) \neq j} - (|\mathcal{J}|-1)c_{\max} + \sum_{j \in \mathcal{J}} l(y,\widehat{y}_{\leq j}) + \sum_{j \in \mathcal{J}} \tilde{c}_j(x,\widehat{y}_{<j},y)$$
$$\leq \sum_{j \in \mathcal{J}} \left(c_{\max} - l(y,\widehat{y}_{\leq j}) - \tilde{c}_j(x,\widehat{y}_{<j},y)\right)\mathbf{1}_{r(x) \neq j}$$

If $r(x) \neq j$, then $\psi_{\mathrm{ce}}(\boldsymbol{g}(x),j) \geq \log 2$ and $\psi_{\mathrm{mae}}(\boldsymbol{g}(x),j) \geq 0.5$. When $r(x) = j$, $\psi_{\mathrm{ce}}(\boldsymbol{g}(x),j) \geq 0$ and $\psi_{\mathrm{mae}}(\boldsymbol{g}(x),j) \geq 0$. Therefore, $\gamma_{\mathrm{ce}}\psi_{\mathrm{ce}}(\boldsymbol{g}(x),j) \geq \mathbf{1}_{r(x) \neq j}$ and $\gamma_{\mathrm{mae}}\psi_{\mathrm{mae}}(\boldsymbol{g}(x),j) \geq \mathbf{1}_{r(x) \neq j}$ where $\gamma_{\mathrm{ce}} = \log 0.5$ and $\gamma_{\mathrm{mae}} = 2$. So then,

$$\mathcal{L}^{\mathrm{OneTime}}(h,r,x,y) \leq \sum_{j \in \mathcal{J}} \left(c_{\max} - l(y,\widehat{y}_{\leq j}) - \tilde{c}_j(x,\widehat{y}_{<j},y)\right)\mathbf{1}_{r(x) \neq j}$$
$$\leq \gamma \sum_{j \in \mathcal{J}} \left(c_{\max} - l(y,\widehat{y}_{\leq j}) - \tilde{c}_j(x,\widehat{y}_{<j},y)\right)\psi(\boldsymbol{g}(x),j)$$

Therefore, $\gamma\mathcal{L}^{\psi}(h,r,x,y) \geq \mathcal{L}^{\mathrm{OneTime}}(h,r,x,y)$

$\square$

### B.3 Proof of Lemma B.3

**Lemma B.3.** *Let $\tilde{c}_j(x, \widehat{y}_{<j}, y) \le \bar{C} < \infty$, $l(y, \widehat{y}_j^h) \le \bar{l} < \infty$ for all $j \in \mathcal{J}$, and $\psi$ be a multiclass surrogate loss that satisfies the following inequality for any distribution over $\mathcal{X} \times \mathcal{J}$ and any measurable function $f$:*

$$\mathcal{R}_{multi\ 0-1}(f) - \mathcal{R}^*_{multi\ 0-1} \le \Gamma\left(\mathcal{R}_{multi\ \psi}(f) - \mathcal{R}^*_{multi\ \psi}\right)$$

*where $\mathcal{R}_{multi\ 0-1}$ is the multiclass 0-1 risk and $\Gamma : \mathbb{R}^+ \to \mathbb{R}^+$ is a non-decreasing concave function. If there exists $i, j \in \mathcal{J}$ such that $\left| l(y, \widehat{y}_i^h) + \tilde{c}_i(x, \widehat{y}_{<i}, y) - l(y, \widehat{y}_j^h) - \tilde{c}_j(x, \widehat{y}_{<j}, y) \right| > \Delta > 0$, then for any $\boldsymbol{g}$ over all measurable functions and for any $h$, the following inequality holds for any distribution over $\mathcal{X} \times \mathcal{Y}$:*

$$\mathcal{R}_{OneTime}(r, h) - \mathcal{R}^*_{OneTime}(h) \le \tilde{\Gamma}\left(\mathcal{R}_\psi(r, h) - \mathcal{R}^*_\psi(h)\right)$$

*where $\tilde{\Gamma}(z) = (|\mathcal{J}| - 1)\left(\bar{C} + \bar{l}\right)\Gamma\left(\frac{z}{\Delta}\right)$, $\mathcal{R}^*_{OneTime}(h) = \inf_{r \in \mathcal{M}} \mathcal{R}_{OneTime}(r, h)$, and $\mathcal{R}^*_\psi(h) = \inf_r \mathcal{R}_\phi(r, h)$*

*Proof.* Let $C_{\text{OneTime}}(r, h, x) = \mathbb{E}_{y \sim \mathcal{P}_{Y|X=x}}\left[\mathcal{L}^{\text{OneTime}}(h, r, x, y)\right]$ be the pointwise $\mathcal{L}^{\text{OneTime}}$- risk for a given $x$ and predictor $h$ and let $C^*_{\text{OneTime}}(h, x) = \inf_r C_{\text{OneTime}}(r, h, x)$ be the optimal pointwise $\mathcal{L}^{\text{OneTime}}$- risk for a given $x$ and predictor $h$. Similarly, let $C_\psi(r, h, x) = \mathbb{E}_{y \sim \mathcal{P}_{Y|X=x}}\left[\mathcal{L}_\psi^{\text{OneTime}}(h, r, x, y)\right]$ be the pointwise $\mathcal{L}^\psi$- risk for a given $x$ and predictor $h$ and $C^*_\psi(h, x) = \inf_r C_\psi(r, h, x)$ be the optimal pointwise $\mathcal{L}^\psi$- risk for a given $x$ and predictor $h$. For any measurable function $\boldsymbol{g}$

$$C_{\text{OneTime}}(r, h, x) - C^*_{\text{OneTime}}(h, x)$$

$$= \mathbb{E}_{y \sim \mathcal{P}_{Y|X=x}}\left[\sum_{j \in \mathcal{J}}\left(l(y, \widehat{y}_{\le j}) + \tilde{c}_j(x, \widehat{y}_{<j}, y)\right)\mathbf{1}_{r(x)=j}\right]$$

$$- \inf_{\tilde{r}} \mathbb{E}_{y \sim \mathcal{P}_{Y|X=x}}\left[\sum_{j \in \mathcal{J}}\left(l(y, \widehat{y}_{\le j}) + \tilde{c}_j(x, \widehat{y}_{<j}, y)\right)\mathbf{1}_{\tilde{r}(x)=j}\right]$$

$$\overset{(a)}{=} \mathbb{E}_{y \sim \mathcal{P}_{Y|X=x}}\left[\sum_{j \in \mathcal{J}}\left(c_{\max} - l(y, \widehat{y}_{\le j}) - \tilde{c}_j(x, \widehat{y}_{<j}, y)\right)\mathbf{1}_{r(x)\neq j} - (|\mathcal{J}| - 1)c_{\max} + \sum_{j \in \mathcal{J}} l(y, \widehat{y}_{\le j}) + \sum_{j \in \mathcal{J}} \tilde{c}_j(x, \widehat{y}_{<j}, y)\right]$$

$$- \inf_{\tilde{r}} \mathbb{E}_{y \sim \mathcal{P}_{Y|X=x}}\left[\sum_{j \in \mathcal{J}}\left(c_{\max} - l(y, \widehat{y}_{\le j}) - \tilde{c}_j(x, \widehat{y}_{<j}, y)\right)\mathbf{1}_{\tilde{r}(x)\neq j} - (|\mathcal{J}| - 1)c_{\max} + \sum_{j \in \mathcal{J}} l(y, \widehat{y}_{\le j}) + \sum_{j \in \mathcal{J}} \tilde{c}_j(x, \widehat{y}_{<j}, y)\right]$$

$$= \sum_{j \in \mathcal{J}} \mathbb{E}_{y \sim \mathcal{P}_{Y|X=x}}\left[c_{\max} - l(y, \widehat{y}_{\le j}) - \tilde{c}_j(x, \widehat{y}_{<j}, y)\right]\mathbf{1}_{r(x)\neq j} - \inf_{\tilde{r}} \sum_{j \in \mathcal{J}} \mathbb{E}_{y \sim \mathcal{P}_{Y|X=x}}\left[c_{\max} - l(y, \widehat{y}_{\le j}) - \tilde{c}_j(x, \widehat{y}_{<j}, y)\right]\mathbf{1}_{\tilde{r}(x)\neq j}$$

(a) By Lemma B.1. Let $A_j = \mathbb{E}_{y \sim \mathcal{P}_{Y|X=x}}\left[c_{\max} - l(y, \widehat{y}_{\le j}) - \tilde{c}_j(x, \widehat{y}_{<j}, y)\right]$

$$C_{\text{OneTime}}(r, h, x) - C^*_{\text{OneTime}}(h, x)$$

$$= \sum_{j \in \mathcal{J}} A_j \mathbf{1}_{r(x)\neq j} - \inf_{\tilde{r}} \sum_{j \in \mathcal{J}} A_j \mathbf{1}_{\tilde{r}(x)\neq j}$$

$$= \sum_{k \in \mathcal{J}} A_k \left(\sum_{j \in \mathcal{J}} \frac{A_j}{\sum_{k \in \mathcal{J}} A_k} \mathbf{1}_{r(x)\neq j} - \inf_{\tilde{r}} \sum_{j \in \mathcal{J}} \frac{A_j}{\sum_{k \in \mathcal{J}} A_k} \mathbf{1}_{\tilde{r}(x)\neq j}\right)$$

Consider a degenerate distribution on $\tilde{X}$ with a pmf of 1 on $x$. Let $P(\tilde{Y} = j \mid \tilde{X} = x) = \frac{A_j}{\sum_{k \in \mathcal{J}} A_k}$ and $f$ be a measurable function such that $f(x) = r(x)$. Under this distribution,

$$\mathcal{R}_{\text{multi } 0-1}(r) = \sum_{j \in \mathcal{J}} \frac{A_j}{\sum_{k \in \mathcal{J}} A_k} \mathbf{1}_{r(x) \neq j} \qquad \mathcal{R}^*_{\text{multi } 0-1} = \inf_{\tilde{r}} \sum_{j \in \mathcal{J}} \frac{A_j}{\sum_{k \in \mathcal{J}} A_k} \mathbf{1}_{\tilde{r}(x) \neq j}$$

$$\mathcal{R}_{\text{multi } \psi}(r) = \sum_{j \in \mathcal{J}} \frac{A_j}{\sum_{k \in \mathcal{J}} A_k} \psi\left(\boldsymbol{g}(x), j\right) \qquad \mathcal{R}^*_{\text{multi } \psi} = \inf_{\tilde{\boldsymbol{g}}} \sum_{j \in \mathcal{J}} \frac{A_j}{\sum_{k \in \mathcal{J}} A_k} \psi\left(\boldsymbol{g}(x), j\right)$$

From the theorem statement,

$$C_{\text{OneTime}}(r, h, x) - C^*_{\text{OneTime}}(h, x)$$

$$= \sum_{k \in \mathcal{J}} A_k \left( \sum_{j \in \mathcal{J}} \frac{A_j}{\sum_{k \in \mathcal{J}} A_k} \mathbf{1}_{r(x) \neq j} - \inf_{\tilde{r}} \sum_{j \in \mathcal{J}} \frac{A_j}{\sum_{k \in \mathcal{J}} A_k} \mathbf{1}_{\tilde{r}(x) \neq j} \right)$$

$$\leq \sum_{k \in \mathcal{J}} A_k \Gamma \left( \sum_{j \in \mathcal{J}} \frac{A_j}{\sum_{k \in \mathcal{J}} A_k} \psi\left(\boldsymbol{g}(x), j\right) - \inf_{\tilde{\boldsymbol{g}}} \sum_{j \in \mathcal{J}} \frac{A_j}{\sum_{k \in \mathcal{J}} A_k} \psi\left(\tilde{\boldsymbol{g}}(x), j\right) \right)$$

We know that $\sum_{j \in \mathcal{J}} A_j \leq (|\mathcal{J}| - 1) \left(\bar{C} + \bar{l}\right)$ from:

$$\sum_{j \in \mathcal{J}} A_j = \sum_{j \in \mathcal{J}} \mathbb{E}_{y \sim \mathcal{P}_{Y|X=x}} \left[ c_{\max} - l(y, \widehat{y}_{\leq j}) - \tilde{c}_j(x, \widehat{y}_{<j}, y) \right]$$

$$\leq (|\mathcal{J} - 1|) \left(\bar{C} + \bar{l}\right)$$

Since $l(y, \widehat{y}_i^h) + \tilde{c}_i(x, \widehat{y}_{<i}, y) \neq l(y, \widehat{y}_j^h) + \tilde{c}_i(x, \widehat{y}_{<i}, y)$ for a pair $(i, j)$, we can assume that $\sum_{j \in \mathcal{J}} A_j \geq \Delta$ if $\left| l(y, \widehat{y}_i^h) + \tilde{c}_i(x, \widehat{y}_{<i}, y) - l(y, \widehat{y}_j^h) - \tilde{c}_i(x, \widehat{y}_{<i}, y) \right| > \Delta$.

With these bounds,

$$C_{\text{OneTime}}(r, h, x) - C^*_{\text{OneTime}}(h, x)$$

$$\leq \sum_{k \in \mathcal{J}} A_k \Gamma \left( \sum_{j \in \mathcal{J}} \frac{A_j}{\sum_{k \in \mathcal{J}} A_k} \psi\left(\boldsymbol{g}(x), j\right) - \inf_{\tilde{\boldsymbol{g}}} \sum_{j \in \mathcal{J}} \frac{A_j}{\sum_{k \in \mathcal{J}} A_k} \psi\left(\tilde{\boldsymbol{g}}(x), j\right) \right)$$

$$\leq (|\mathcal{J}| - 1) \left(\bar{C} + \bar{l}\right) \Gamma \left( \frac{\sum_{j \in \mathcal{J}} A_j \psi\left(\boldsymbol{g}(x), j\right) - \inf_{\tilde{\boldsymbol{g}}} \sum_{j \in \mathcal{J}} A_j \psi\left(\tilde{\boldsymbol{g}}(x), j\right)}{\Delta} \right)$$

$$= (|\mathcal{J}| - 1) \left(\bar{C} + \bar{l}\right) \Gamma \left( \frac{C_\psi(r, h, x) - C^*_\psi(h, x)}{\Delta} \right)$$

Let $\tilde{\Gamma}(z) = (|\mathcal{J}| - 1) \left(\bar{C} + \bar{l}\right) \Gamma\left(\frac{z}{\Delta}\right)$. Then,

$$\mathcal{R}_{\text{OneTime}}(r, h) - \mathcal{R}^*_{\text{OneTime}}(h) = \mathbb{E}_{x \sim \mathcal{P}_X} \left[ C_{\text{OneTime}}(r, h, x) - C^*_{\text{OneTime}}(h, x) \right]$$

$$\leq \mathbb{E}_{x \sim \mathcal{P}_X} \left[ \tilde{\Gamma} \left( C_\psi(r, h, x) - C^*_\psi(h, x) \right) \right]$$

$$\overset{(a)}{\leq} \tilde{\Gamma} \left( \mathbb{E}_{x \sim \mathcal{P}_X} \left[ C_\psi(r, h, x) - C^*_\psi(h, x) \right] \right)$$

$$\overset{(b)}{=} \tilde{\Gamma} \left( \mathcal{R}_\psi(r, h) - \mathcal{R}^*_\psi(h) \right)$$

(a) by Jensen's inequality since $\tilde{\Gamma}$ is concave, (b) since the infimum is taken over all measurable functions $\mathcal{R}^*_\psi(h) = \mathbb{E}_{x \sim \mathcal{P}_X} \left[ C^*_\psi(h, x) \right]$  □

### B.4 Proof of Theorem 4.1

**Theorem B.4** (Consistency). *Let $\tilde{c}_j(x, \widehat{y}_{<j}, y) \leq \bar{C} < \infty$, $l(y, \widehat{y}_j^h) \leq \bar{l} < \infty$ for all $j \in \mathcal{J}$, and $\psi$ be cross entropy loss $\psi_{ce}$ or mean absolute error $\psi_{mae}$. If there exists $i, j \in \mathcal{J}$ such that $\left| l(y, \widehat{y}_i^h) + \tilde{c}_i(x, \widehat{y}_{<i}, y) - l(y, \widehat{y}_j^h) - \tilde{c}_j(x, \widehat{y}_{<j}, y) \right| > \Delta > 0$, then, for a fixed $h$, $\mathcal{L}^\psi$ is a consistent surrogate for $\mathcal{L}^{OneTime}$*

*Proof.* We know that $\mathcal{R}_{\text{multi } 0-1}(f) - \mathcal{R}^*_{\text{multi } 0-1} \leq \Gamma_{ce}\left(\mathcal{R}_{\text{multi } \psi_{ce}}(f) - \mathcal{R}^*_{\text{multi } \psi_{ce}}\right)$ from Theorem 3.1 of Mao et al. (2023) where $\Gamma_{ce}^{-1}(z) = \frac{1+z}{2}\log(1+z) + \frac{1-z}{2}\log(1-z)$. Similarly, $\mathcal{R}_{\text{multi } 0-1}(f) - \mathcal{R}^*_{\text{multi } 0-1} \leq \Gamma_{mae}\left(\mathcal{R}_{\text{multi } \psi_{mae}}(f) - \mathcal{R}^*_{\text{multi } \psi_{mae}}\right)$ where $\Gamma_{mae}^{-1}(z) = \frac{z}{L+1}$. Both $\Gamma_{ce}^{-1}$ and $\Gamma_{mae}^{-1}$ are non-decreasing convex functions making $\Gamma_{ce}$ and $\Gamma_{mae}$ non-decreasing *concave* functions on $\mathbb{R}^+$ domain.

Let $\Gamma \equiv \Gamma_{ce}$ when $\psi \equiv \psi_{ce}$ and $\Gamma \equiv \Gamma_{mae}$ when $\psi \equiv \psi_{mae}$. By Lemma B.3, $\mathcal{R}_{\text{OneTime}}(r) - \mathcal{R}^*_{\text{OneTime}} \leq \tilde{\Gamma}\left(\mathcal{R}_\psi(r) - \mathcal{R}^*_\psi\right)$ where $\tilde{\Gamma}(z) = (|\mathcal{J}| - 1)\left(\bar{C} + \bar{l}\right)\Gamma\left(\frac{z}{\Delta}\right)$. If $\lim_{n\to\infty} \mathcal{R}_\psi(r_n) - \mathcal{R}^*_\psi = 0$, then

$$
\begin{aligned}
\lim_{n\to\infty} \mathcal{R}_{\text{Point}}(r_n) - \mathcal{R}^*_{\text{Point}} &\leq \lim_{n\to\infty} \tilde{\Gamma}\left(\mathcal{R}_\psi(r_n) - \mathcal{R}^*_\psi\right) \\
&\overset{(a)}{=} (|\mathcal{J}| - 1)\left(\bar{C} + \bar{l}\right)\Gamma\left(\lim_{n\to\infty} \frac{\mathcal{R}_\phi(r_n) - \mathcal{R}^*_\phi}{\Delta}\right) \\
&= (|\mathcal{J}| - 1)\left(\bar{C} + \bar{l}\right)\Gamma(0) \\
&\overset{(b)}{=} 0
\end{aligned}
$$

(a) by continuity of $\Gamma$ at 0 and (b) since $\Gamma_{ce}(0) = 0$ and $\Gamma_{mae}(0) = 0$

$\square$

### B.5 Proof of Theorem 4.2

**Theorem B.5.** *Suppose $r_n \in \mathcal{F} = \{x \to \arg\min_{j \in \mathcal{J}} g_j(x) : g_j \in \mathcal{G}_j\}$ where $\mathcal{G}_j$ consists of functions having a range of $(-M, M)$ with $M \geq 0$, $\tilde{c}_j(x, \widehat{y}_{<j}, y) \leq \bar{C}$, and $l(y, \widehat{y}_j^h) \leq \bar{l} < \infty$. If $\psi$ is $\rho$-lipschitz continuous with respect to $\boldsymbol{g}(x)$ and there exists $u_\psi(M)$ such that $\psi(\boldsymbol{g}(x), j) \leq u_\psi(M) < \infty$ for all $j \in \mathcal{J}$ and $r \in \mathcal{F}$, then with probability $1 - \delta$ for a fixed $h$, the following upper bound holds for the empirical risk minimizer $\widehat{r}_n$ with respect to $\mathcal{L}^\psi$:*

$$
\mathcal{R}_\psi(\widehat{r}_n, h) - \mathcal{R}^*_\psi(\mathcal{F}, h) \leq 2\sqrt{2}q(L)\rho \sum_{j \in \mathcal{J}} \mathfrak{R}_n(\mathcal{G}_j) + u_\psi(M)q(L)\sqrt{\frac{2\log\left(\frac{2}{\delta}\right)}{n}}
$$

*where $\mathcal{R}^*_\psi(\mathcal{F}, h) = \inf_{r \in \mathcal{F}} \mathcal{R}_\psi(r, h)$, $q(L) = 2(|\mathcal{J}| - 1)(\bar{C} + \bar{l})$, and $\mathfrak{R}_n(\mathcal{G}_j)$ is the Rademacher Complexity of $\mathcal{G}_j$.*

*Proof.* Let $G_n = \sup_{r \in \mathcal{F}}\left[\mathcal{R}_\psi(r) - \widehat{\mathcal{R}}_\psi(r)\right]$, $G_{n'} = \sup_{r \in \mathcal{F}}\left[\widehat{\mathcal{R}}_\psi(r) - \mathcal{R}_\psi(r)\right]$, and $\mathcal{B} = \{(x, y) \mapsto \mathcal{L}^\psi(h, r, x, y) : r \in \mathcal{F}\}$.

If $z^i = (x^i, y^i)$, $G_n = g(z^1, \cdots, z^n)$.

$$
\left| g(z^1, \cdots, z^i, \cdots, z^n) - g(z^1, \cdots, z^{i'}, \cdots, z^n) \right| \leq \frac{1}{n_c} \left| \sup_{r \in \mathcal{F}} \mathcal{L}^\psi \left( h, r, x^i, y^i \right) - \mathcal{L}^\psi \left( h, r, x^{i'}, y^{i'} \right) \right|
$$

$$
\leq \frac{1}{n} \sup_{r \in \mathcal{F}} \sum_{j \in \mathcal{J}} \left| \left( c^i_{\max} - l(y^i, \widehat{y}^i_{\leq j}) - \tilde{c}_j(x^i, \widehat{y}^i_{<j}, y^i) \right) \psi(\boldsymbol{g}(x^i), y^i) \right|
$$

$$
+ \left| \left( c^{i'}_{\max} - l(y^{i'}, \widehat{y}^{i'}_{\leq j}) - \tilde{c}_j(x^{i'}, \widehat{y}^{i'}_{<j}, y^{i'}) \right) \psi(\boldsymbol{g}(x^{i'}), y^{i'}) \right|
$$

$$
\leq \frac{1}{n} u_\psi(M) 2(|\mathcal{J}| - 1)(\bar{C} + \bar{l})
$$

$$
= \frac{1}{n} u_\psi(M) q(L)
$$

where $q(L) = 2(|\mathcal{J}| - 1)(\bar{C} + \bar{l})$. Using this boundedness property, we can apply McDiarmid's inequality

$$
P\left( G_n > \frac{\epsilon}{2} \right) \leq \exp \left\{ \frac{-2 \left( \frac{\epsilon}{2} - \mathbb{E}[G_n] \right)^2}{\sum_{i=1}^{n_c} \frac{u_\psi(M)^2 q(L)^2}{n^2}} \right\}
$$

$$
\leq \exp \left\{ \frac{-2n \left( \frac{\epsilon}{2} - 2\mathfrak{R}_n(\mathcal{B}) \right)^2}{u_\psi(M)^2 q(L)^2} \right\}
$$

$$
\overset{\text{def}}{=} \frac{\delta}{2}
$$

Similarly,

$$
P\left( G'_n > \frac{\epsilon}{2} \right) \leq \exp \left\{ \frac{-2n \left( \frac{\epsilon}{2} - 2\mathfrak{R}_n(\mathcal{B}) \right)^2}{u_\psi(M)^2 q(L)^2} \right\} = \frac{\delta}{2}
$$

Then $\epsilon = 4\mathfrak{R}_n(\mathcal{B}) + u_\psi(M) q(L) \sqrt{\frac{2 \log\left(\frac{2}{\delta}\right)}{n}}$

$$
P\left( \mathcal{R}_\psi(\widehat{r}_n) - \mathcal{R}^*_\psi(\mathcal{F}) > \epsilon \right) \leq P\left( \sup_{r \in \mathcal{F}} \left| \mathcal{R}_\psi(r) - \widehat{\mathcal{R}}_\psi(r) \right| > \frac{\epsilon}{2} \right)
$$

$$
\leq P\left( G_n > \frac{\epsilon}{2} \right) + P\left( G'_n > \frac{\epsilon}{2} \right)
$$

$$
\leq \delta
$$

With at least probability $1 - \delta$, $\mathcal{R}_\psi(\widehat{r}_n, h) - \mathcal{R}^*_\psi(\mathcal{F}, h) \leq 4\mathfrak{R}_n(\mathcal{B}) + u_\psi(M) q(L) \sqrt{\frac{2 \log\left(\frac{2}{\delta}\right)}{n}}$.

We can further bound $\mathfrak{R}_n(\mathcal{B})$ by $\mathfrak{R}_n(\mathcal{F}_j)$ if we show that $\mathcal{L}^\psi$ is Lipschitz continuous with respect to $\boldsymbol{g}$

$$\left\|\mathcal{L}^{\psi}(h, r^1, x, y) - \mathcal{L}^{\psi}(h, r^2, x, y)\right\|_2$$

$$= \left\|\sum_{j \in \mathcal{J}} (c_{\max} - l(y, \widehat{y}_{\leq j}) - \tilde{c}_j(x, \widehat{y}_{<j}, y)) \psi(\boldsymbol{g}^1(x), j) - \sum_{j \in \mathcal{J}} (c_{\max} - l(y, \widehat{y}_{\leq j}) - \tilde{c}_j(x, \widehat{y}_{<j}, y)) \psi(\boldsymbol{g}^2(x), j)\right\|_2$$

$$\leq \sum_{j \in \mathcal{J}} (c_{\max} - l(y, \widehat{y}_{\leq j}) - \tilde{c}_j(x, \widehat{y}_{<j}, y)) \left\|\psi(\boldsymbol{g}^1(x), j) - \psi(\boldsymbol{g}^2(x), j)\right\|_2$$

$$\overset{(a)}{=} \sum_{j \in \mathcal{J}, j \neq j_{\max}} (c_{\max} - l(y, \widehat{y}_{\leq j}) - \tilde{c}_j(x, \widehat{y}_{<j}, y)) \left\|\psi(\boldsymbol{g}^1(x), j) - \psi(\boldsymbol{g}^2(x), j)\right\|_2$$

$$\overset{(b)}{\leq} \sum_{j \in \mathcal{J}, j \neq j_{\max}} (\bar{C} + \bar{l}) \rho \left\|\boldsymbol{g}^1(x) - \boldsymbol{g}^2(x)\right\|_2$$

$$= (\mathcal{J} - 1) (\bar{C} + \bar{l}) \rho \left\|\boldsymbol{g}^1(x) - \boldsymbol{g}^2(x)\right\|_2$$

(a) where $j_{\max} = \arg\max_{j \in \mathcal{J}} l(y, \widehat{y}_{\leq j}) + \tilde{c}_j(x, \widehat{y}_{<j}, y)$, (b) by lipschitz continuity of $\psi$.

By the extension of the Talagrand's Contraction Lemma Maurer (2016),

$$\mathfrak{R}_n(\mathcal{B}) = \mathbb{E}_{\sigma_1, \ldots, \sigma_n, \{z^i\}_{i=1}^n \sim \mathcal{P}_{\mathcal{X} \times \mathcal{Y}}} \left[\sup_{b \in \mathcal{B}} \frac{1}{n} \sum_{i=1}^n \sigma_i b(z^i)\right]$$

$$\leq (|\mathcal{J}| - 1) (\bar{C} + \bar{l}) \rho \sqrt{2} \mathbb{E}_{\sigma, \{z^i\}_{i=1}^n \sim \mathcal{P}_{\mathcal{X} \times \mathcal{Y}}} \left[\sup_{\boldsymbol{g} \in \mathcal{G}_1 \times \ldots \times \mathcal{G}_{L+1}} \frac{1}{n} \sum_{i=1}^n \sum_{j \in \mathcal{J}} \sigma_{i,j} g_j(z^i)\right]$$

$$\leq \frac{q(L)\rho}{\sqrt{2}} \sum_{j \in \mathcal{J}} \mathbb{E}_{\sigma, \{z^i\}_{i=1}^n \sim \mathcal{P}_{\mathcal{X} \times \mathcal{Y}}} \left[\sup_{g_j \in \mathcal{G}_j} \frac{1}{n} \sum_{i=1}^n \sigma_{i,j} g_j(z^i)\right]$$

$$\leq \frac{q(L)\rho}{\sqrt{2}} \sum_{j \in \mathcal{J}} \mathfrak{R}_n(\mathcal{G}_j)$$

$\square$

## B.6 Proof of Corollary B.6

**Corollary B.6.** *Suppose $r_n \in \mathcal{F} = \{x \to \arg\min_{j \in \mathcal{J}} g_j(x) : g_j \in \mathcal{G}_j\}$ where $\mathcal{G}_j$ consists of functions having a range of $(-M, M)$ with $M \geq 0$, $\tilde{c}_j(x, \widehat{y}_{<j}, y) \leq \bar{C}$, and $l(y, \widehat{y}_j^h) \leq \bar{l} < \infty$. If $\psi$ is $\psi_{ce}$ or $\psi_{mae}$, then with probability $1 - \delta$, for a fixed $h$, the excess risk of with respect to $\mathcal{L}^{OneTime}$ holds for the empirical risk minimizer $\widehat{r}_n$ with respect to $\mathcal{L}^{\psi}$*

$$\mathcal{R}_{OneTime}(\widehat{r}_n, h) - \mathcal{R}_{OneTime}^*(h) \leq \tilde{\Gamma}\left(2q(L)\rho\sqrt{2}\sum_{j \in \mathcal{J}} \mathfrak{R}_n(\mathcal{G}_j) + u_\psi(M)q(L)\sqrt{\frac{2\log\left(\frac{2}{\delta}\right)}{n}} + \mathcal{A}_\psi(\mathcal{F}, h)\right)$$

*where $\tilde{\Gamma} : \mathbb{R}^+ \to \mathbb{R}^+$, $q(L) = 2(|\mathcal{J}| - 1)(\bar{C} + \bar{l})$, $\rho$ is the lipschitz norm of $\psi$, and $\mathcal{A}_\psi(\mathcal{F}, h) = \mathcal{R}_\psi^*(\mathcal{F}, h) - \mathcal{R}_\psi^*(h)$ is the approximation error.*

*Proof.* Excess risk can be decomposed into estimation error and approximation error

$$\mathcal{R}_\psi(r, h) - \mathcal{R}_\psi^* = \underbrace{\mathcal{R}_\psi(r, h) - \mathcal{R}_\psi^*(h)(\mathcal{F}, h)}_{\text{Estimation Error}} + \underbrace{\mathcal{R}_\psi^*(\mathcal{F}, h) - \mathcal{R}_\psi^*(h)}_{\text{Approximation Error}}$$

We know that $\psi_{ce}, \psi_{mae}$ are upper bounded by $u_{\psi_{ce}}(M), u_{\psi_{mae}}(M)$ repsectively. We also know that the lipschitz norm for $\psi_{ce}$ $\rho_{ce}$ is $\sqrt{2}$ and the lipschitz norm for $\psi_{mae}$ $\rho_{mae}$ is 2

From Theorem 4.2, we have an upper bound on the estimation error and so using that, we have the following inequality

$$\mathcal{R}_{\psi}(r, h) - \mathcal{R}_{\psi}^{*}(h) \leq 2q(L)\rho\sqrt{2} \sum_{j \in \mathcal{J}} \mathfrak{R}_n\left(\mathcal{G}_j\right) + u_{\psi}(M)q(L)\sqrt{\frac{2\log\left(\frac{2}{\delta}\right)}{n}} + \mathcal{A}_{\psi}\left(\mathcal{F}, h\right)$$

$\mathcal{R}_{\text{multi } 0-1}(f) - \mathcal{R}_{\text{multi } 0-1}^{*} \leq \Gamma_{\text{ce}}\left(\mathcal{R}_{\text{multi } \psi_{\text{ce}}}(f) - \mathcal{R}_{\text{multi } \psi_{\text{ce}}}^{*}\right)$ from Theorem 3.1 of Mao et al. (2023) where $\Gamma_{\text{ce}}^{-1}(z) = \frac{1+z}{2}\log(1 + z) + \frac{1-z}{2}\log(1 - z)$. Similarly, $\mathcal{R}_{\text{multi } 0-1}(f) - \mathcal{R}_{\text{multi } 0-1}^{*} \leq \Gamma_{\text{mae}}\left(\mathcal{R}_{\text{multi } \psi_{\text{mae}}}(f) - \mathcal{R}_{\text{multi } \psi_{\text{mae}}}^{*}\right)$ where $\Gamma_{\text{mae}}^{-1}(z) = \frac{z}{L+1}$. Both $\Gamma_{\text{ce}}^{-1}$ and $\Gamma_{\text{mae}}^{-1}$ are non-decreasing convex functions making $\Gamma_{\text{ce}}$ and $\Gamma_{\text{mae}}$ non-decreasing *concave* functions on $\mathbb{R}^{+}$ domain.

The proof is complete by using Lemma B.3

$\square$

## C   Baseline Details

For XSUM and TSP, let $p_h(\widehat{y}_j \mid \widehat{y}_{<j}, x)$ be the predictive distribution of the $j^{\text{th}}$ token conditioned on the leftward context $\widehat{y}_{<j}$ and input $x$. The confidence measure would then be $-\log p_h(\widehat{y}_j \mid \widehat{y}_{<j}, x)$ and the predictive entropy would be $H(\widehat{Y}_j \mid \widehat{y}_{<j}, x) = -\sum_{v \in \mathcal{V}} p_h(v \mid \widehat{y}_{<j}, x) \log p_h(v \mid \widehat{y}_{<j}, x)$. For MWP, if we assume that there are dropout layers in the predictor network, we can use the Monte Carlo Variance Gal & Ghahramani (2016) or $\text{Var}_h(v \mid y_{<j}, x)$ across $T$ inference passes as a confidence measure.

For a given threshold $\tau$, the whole deferral decision rule is determined by evaluating $r(x) > \tau$. The baselines offer various ways of modeling $r$.

- `ChowSum` - For TSP and XSUM, $r_{\text{ChowSum}}(x) = -\sum_{j=1}^{L} \log p_h(\widehat{y}_j \mid \widehat{y}_{<j}, x)$ and for MWP, $r_{\text{ChowSum}}(x) = \sum_{j=1}^{L} \text{Var}_h(v \mid y_{<j}, x)$

- `ChowMean` - For TSP and XSUM, $r_{\text{ChowMean}}(x) = -\frac{1}{L}\sum_{j=1}^{L} \log p_h(\widehat{y}_j \mid \widehat{y}_{<j}, x)$ and for MWP, $r_{\text{ChowSum}}(x) = \frac{1}{L}\sum_{j=1}^{L} \text{Var}_h(v \mid y_{<j}, x)$

- `ChowQuantile` at the level $\alpha$ - For TSP and XSUM, $r_{\text{ChowQuantile}}^{\alpha}(x) = \text{Quantile}_\alpha\left(\{-\log p_h(\widehat{y}_j \mid \widehat{y}_{<j}, x)\}_{j=1}^{L}\right)$ and for MWP, $r_{\text{ChowSum}}(x) = \text{Quantile}_\alpha\left(\{\text{Var}_h(v \mid y_{<j}, x)\}_{j=1}^{L}\right)$

- `WholeModelScore` - $r(x)$ is a trained model with the inputs for the rejector are the various quantiles of the confidence scores. The training details can be found in Table 2.

- `WholeModelEmbed` - $r(x)$ is trained just like `WholeModelScore` but the inputs for the rejector are the inputs to the predictor along with the confidence scores. The training details can be found in Table 2.

| Task/Hyperparameters | TSP-WholeModelEmbed | TSP-WholeModelScore | XSUM-WholeModelEmbed | XSUM-WholeModelScore | MWP-WholeModelEmbed | MWP-WholeModelScore |
|---|---|---|---|---|---|---|
| Learning rate | $1 \times 10^{-4}$ | $1 \times 10^{-4}$ | $1 \times 10^{-4}$ | $1 \times 10^{-4}$ | $1 \times 10^{-4}$ | $1 \times 10^{-4}$ |
| Early Stopping (patience, delta) | $20, 1 \times 10^{-4}$ | $20, 1 \times 10^{-4}$ | $20, 1 \times 10^{-4}$ | $20, 1 \times 10^{-4}$ | $3, 1 \times 10^{-4}$ | $3, 1 \times 10^{-4}$ |
| Architecture | MLP | MLP | MLP | MLP | MLP | MLP |
| Hidden layers | 1 | 4 | 1 | 4 | 1 | 4 |
| Hidden units | 8 | 32 | 8 | 32 | 8 | 32 |
| Dropout | 0.0 | 0.0 | 0.0 | 0.0 | 0.0 | 0.0 |
| Epochs | 500 | 500 | 500 | 500 | 500 | 500 |
| Training Samples | 800 | 800 | 2400 | 2400 | 105120 | 105120 |
| Gradient Clipping Norm | 1.0 | 1.0 | 1.0 | 1.0 | 1.0 | 1.0 |
| Weight Decay | 0.005 | 0.005 | 0.005 | 0.005 | 0.005 | 0.005 |

Table 2: Hyperparameter settings for WholeModel baselines

Similarly, for token-wise deferral, if $r_j(x, \widehat{y}_{<j})$ exceeds threshold $\tau$, the expert is called to predict the next token. $r_j$ can be modeling in the following ways:

- `TokenwiseScore`: For TSP and XSUM, $r_j^{\mathrm{score}}(x, \widehat{y}_{<j}) = -\log p_h(\widehat{y}_j \mid \widehat{y}_{<j}, x)$ and for MSP, $r_j^{\mathrm{softmax}}(x, \widehat{y}_{<j}) = \mathrm{Var}_h(v \mid y_{<j}, x)$

- `TokenwiseEntropy`: $r_j^{\mathrm{entropy}}(x, \widehat{y}_{<j}) = H(\widehat{Y}_j \mid \widehat{y}_{<j}, x)$. This only applies to XSUM and TSP.

Since $r(x) = \arg\max_{j \in [1, L+1]} g_j(x)$ for one-time deferral, we consider the following the scoring functions $g_j$:

- `OneTimeScore`: For TSP and XSUM, $g_j^{\mathrm{score}}(x) = -\log p_h(\widehat{y}_j \mid \widehat{y}_{<j}, x)$ and for MSP, $g_j^{\mathrm{score}}(x) = \mathrm{Var}_h(v \mid y_{<j}, x)$

- `OneTimeEntropy`: $g_j^{\mathrm{entropy}}(x, \widehat{y}_{<j}) = H(\widehat{Y}_j \mid \widehat{y}_{<j}, x)$. This only applies to XSUM and TSP.

## D  Training Details and Hyperparameters

All the models were implemented using PyTorch and all the models were trained on an Nvidia A40 GPU. They were trained using Early stopping and Table 3 contains all hyperparameter details.

| Task/Hyperparameters | TSP-OneTime | XSUM-Tokenlevel | XSUM-OneTime | MWP-Tokenlevel | MWP-OneTime |
|---|---|---|---|---|---|
| Learning rate | 5e-4 | 1e-3 | 5e-4 | 5e-4 | 5e-4 |
| Early Stopping (patience, delta) | 20, 1e-4 | 7, 1e-4 | 20, 1e-4 | 7, 1e-4 | 20, 1e-4 |
| Architecture | MLP | LSTM | MLP | LSTM | MLP |
| Hidden layers | 1 | 3 | 1 | 2 | 1 |
| Hidden units | 8 | 128 | 8 | 64 | 8 |
| Dropout | 0.2 | 0.4 | 0.2 | 0.4 | 0.2 |
| Epochs | 200 | 100 | 200 | 100 | 200 |
| Training Samples | 2400 | 800 | 2400 | 70080 | 70080 |
| Gradient Clipping Norm | 1.0 | 1.0 | 1.0 | 1.0 | 1.0 |
| Weight Decay | 0.005 | 0.001 | 0.005 | 0.001 | 0.005 |

Table 3: Hyperparameter settings

**TSP**: Using the RL4CO library Berto et al. (2023), TSP graphs with 50 nodes were generated by sampling the node coordinates from a standard normal distribution. These samples were fed into a PointerNet (predictor) trained with 10000 samples. The node embeddings, graph context, and log scores from the PointerNet served as inputs for the One-Time rejector. This rejector only took a maximum of 10 minutes to train.

**XSUM**: The model was trained without teacher forcing. It took about 5 hours to train. The token level rejector $r$ recieves the encoder hidden states from the predictor as hidden states and the decoder hidden states as input. This continues in an autoregressive fashion. We defer the reader to Appendix H for ablation studies on various training strategies for the tokenlevel rejector.

The OneTime rejector was trained on a dataset with summaries containing upto 64 words and it took a couple of minutes to train after a dataset of all the varying mixtures of predictions was created (this took a couple of hours to make). This rejector takes the encoder hidden states, decoder hidden states, and log scores from the predictor as input.

**MWP**: The token-level rejector took about 5 hour to train. e token level rejector $r$ recieves the encoder hidden states from the predictor as hidden states and the decoder hidden states as input. This continues in an autoregressive fashion. The OneTime rejector was trained on a dataset containing all the varying mixtures of predictions which took a couple of minutes to generate. This rejector takes the encoder hidden states, decoder hidden states, and monte carlo variance estimates from the predictor as input.

## E  Paired Significance Results

See Table 4 for paired t-test results.

Table 4: One-sided paired $t$-test $p$-values for AUDC vs. the random rejector (testing whether AUDC is lower than random; best OneTimeModel config per task; $^*p < 0.05$, $^{**}p < 0.01$, $^{***}p < 0.001$.

| Method | TSP | XSUM | MWP |
|---|---|---|---|
| Random | – | – | – |
| Optimal | $< 0.001^{***}$ | $< 0.001^{***}$ | $< 0.001^{***}$ |
| Whole Deferrals | | | |
| ChowMean | $< 0.001^{***}$ | 0.641 | 0.339 |
| ChowSum | $< 0.001^{***}$ | 0.387 | 0.339 |
| ChowQuantile 0 | 0.443 | 0.701 | 0.409 |
| ChowQuantile 0.4 | $< 0.001^{***}$ | 0.665 | 0.409 |
| ChowQuantile 0.8 | $< 0.001^{***}$ | 0.594 | 0.409 |
| ChowQuantile 1.0 | $< 0.001^{***}$ | 0.461 | 0.409 |
| WholeModelEmbed | 0.973 | 0.509 | 0.375 |
| WholeModelScore | 0.460 | 0.356 | 0.616 |
| One-time Deferrals | | | |
| OneTimeModel | $< 0.001^{***}$ | $0.006^{**}$ | $0.014^*$ |
| OneTimeScore | 1.000 | 1.000 | 0.520 |
| OneTimeEntropy | 1.000 | 1.000 | – |
| Token-level Deferrals | | | |
| TokenwiseModel | – | $< 0.001^{***}$ | $< 0.001^{***}$ |
| TokenwiseScore | – | $< 0.001^{***}$ | 1.000 |
| TokenwiseEntropy | – | $< 0.001^{***}$ | – |

## F  Effects of $\alpha_j$

In experiments, we determine the cost of deferring a complete prediction and call it $\alpha_1$. Then, we set $\alpha_j = \frac{L-j+1}{L}\alpha_1$, making the cost of deferring a token $\frac{\alpha_1}{L}$ or $\alpha_L$. We vary $\alpha_1$ and Figure 3 shows how the costs affect cost-accuracy tradeoffs. For both experiments, extreme values for cost often result in poor cost-accuracy tradeoff. For TSP, the results are best when $\alpha_L = 0.01$ and for XSUM, it is either $\alpha_L = 0.01$ or $\alpha_L = 0.015$.

As a general rule of thumb, we propose that $\alpha_1$ should be determined by the median difference between the expert accuracy and predictor accuracy on the train set.

## G  AUDC and Percent Improvement for Size of $\mathcal{J}$ experiment

In support of Figure 2, Tables 5 to 7 report the actual values of AUDC and percent improvement in AUDC over the random baseline for each size of $\mathcal{J}$ considered for TSP, XSUM, and MWP.

## H  Token-level Rejector Ablation

$\mathcal{L}^{\text{Token}}$ bears resemblance to a weighted multi-label classification loss function. Specifically, the token-level rejector follows a classifier chain approach Read et al. (2021). Classifier chains is a multi-label classification (MLC) approach that involves training $L$ binary classifiers in a pre-specified order with the $j^{\text{th}}$ binary classifier taking the feature vector and the previous $j-1$ labels as input, forming a chain of predictors. At inference time, the test features are fed into the chain such that the previous label predictions are appended to the features for the next label classifier. Our token-level rejector $r_j$ implicitly makes these chained predictions by taking in $\widehat{y}_{<j}$ which is informed by the previous $(j-1)$ rejectors' rejection decision.

With this comparison established, model architectures and training techniques from classifier chain literature can be utilized. Recurrent classifier chains Nam et al. (2017) have emerged as a popular choice for MLC

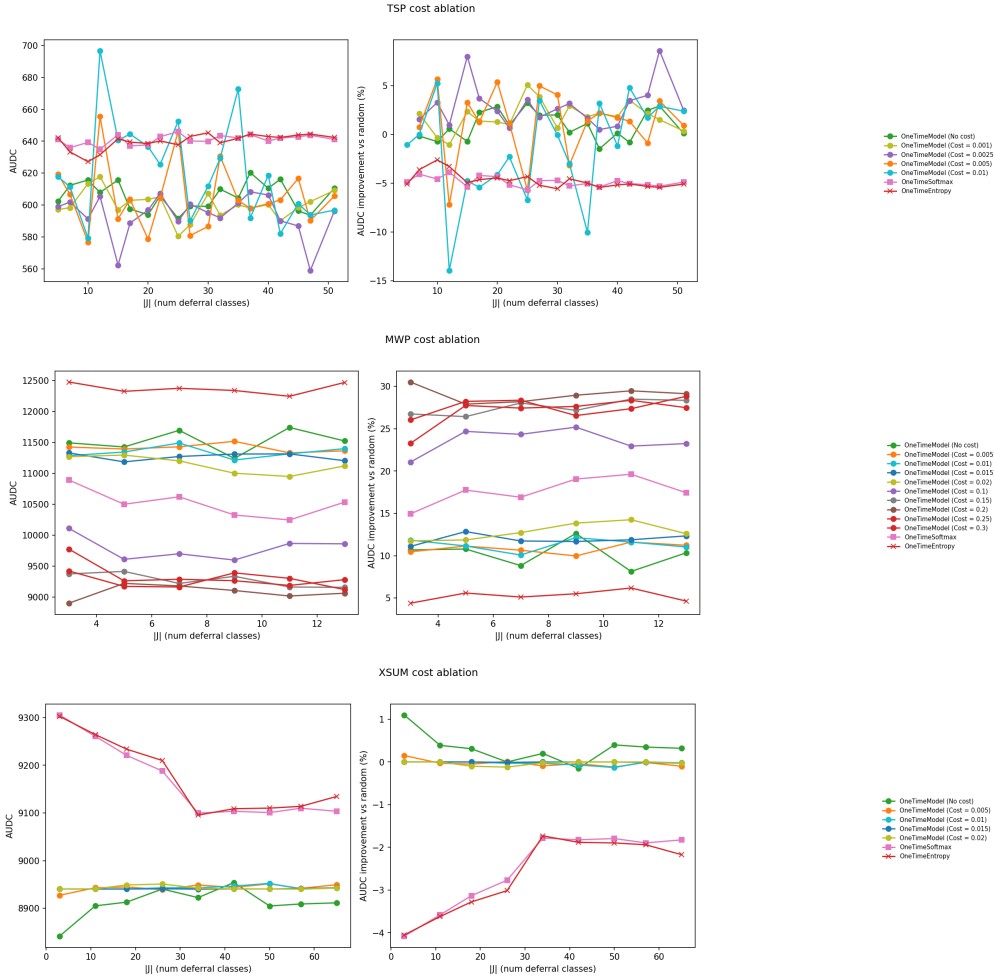

Figure 3: Plots of Size of $\mathcal{J}$ against AUDC and percentage improvement in AUDC relative to the random baseline for TSP (top), MWP (middle), and XSUM (bottom) for models trained with different $\alpha_1$ value. The costs displayed in the legend refer to the deferral cost per token or $\alpha_L$

| Size of $\mathcal{J}$ | OneTimeModel (Ours) | | OneTimeSoftmax | | OneTimeEntropy | |
|---|---|---|---|---|---|---|
| | AUDC | % Improvement | AUDC | % Improvement | AUDC | % Improvement |
| 5 | 611.32(16.83) | -0.00(0.00) | 641.09(16.63) | -4.89(1.97) | 642.24(17.21) | -5.08(2.24) |
| 7 | 588.24(18.15) | 3.78(1.30) | 635.97(16.00) | -4.07(2.48) | 633.29(19.90) | -3.61(2.43) |
| 10 | 576.68(16.51) | 5.66(0.96) | 639.25(18.51) | -4.58(1.95) | 627.14(15.61) | -2.62(2.07) |
| 12 | 600.84(18.63) | 1.71(1.36) | 634.94(20.37) | -3.87(2.16) | 631.52(21.43) | -3.30(2.03) |
| 15 | 562.37(13.36) | 7.98(1.72) | 644.07(19.64) | -5.37(2.12) | 641.51(19.75) | -4.95(2.26) |
| 17 | 588.68(14.72) | 3.68(1.79) | 636.99(20.16) | -4.21(2.15) | 639.46(17.49) | -4.64(2.65) |
| 20 | 578.63(18.25) | 5.35(1.31) | 637.72(19.57) | -4.33(1.95) | 638.56(21.21) | -4.46(1.87) |
| 22 | 595.86(17.89) | 2.53(1.37) | 642.83(21.24) | -5.16(2.21) | 640.18(18.82) | -4.75(2.70) |
| 25 | 580.44(18.68) | 5.06(1.32) | 645.97(18.15) | -5.69(1.96) | 637.76(22.03) | -4.32(1.89) |
| 27 | 580.88(16.75) | 4.97(1.41) | 640.16(18.04) | -4.74(2.19) | 643.00(17.58) | -5.21(2.32) |
| 30 | 586.22(15.53) | 4.10(0.93) | 639.88(12.97) | -4.71(2.20) | 645.25(17.56) | -5.57(1.84) |
| 32 | 591.92(17.11) | 3.16(1.65) | 643.54(19.17) | -5.28(2.02) | 639.01(20.53) | -4.55(2.61) |
| 35 | 595.60(18.13) | 2.58(0.64) | 642.01(19.30) | -5.03(2.04) | 641.85(21.87) | -4.99(1.86) |
| 37 | 591.88(20.38) | 3.18(1.88) | 644.14(17.62) | -5.39(2.17) | 644.46(19.30) | -5.43(2.03) |
| 40 | 600.22(17.67) | 1.81(1.48) | 640.13(13.40) | -4.75(1.78) | 642.89(17.90) | -5.18(1.73) |
| 42 | 582.08(17.21) | 4.78(1.39) | 641.95(19.30) | -5.03(2.34) | 642.35(18.94) | -5.08(1.51) |
| 45 | 584.97(14.70) | 4.29(1.83) | 642.79(15.79) | -5.17(1.72) | 643.90(17.49) | -5.34(1.48) |
| 47 | 558.99(22.86) | 8.57(2.43) | 643.72(18.62) | -5.31(1.86) | 644.48(18.28) | -5.43(1.62) |
| 51 | 596.17(18.47) | 2.48(0.84) | 641.10(18.99) | -4.89(2.03) | 642.36(18.45) | -5.09(1.64) |

Table 5: AUDC scores and Percent improvement over the random baseline in TSP of various sizes of $\mathcal{J}$ with standard deviations in brackets

| Size of $\mathcal{J}$ | OneTimeModel (Ours) | | OneTimeSoftmax | | OneTimeEntropy | |
|---|---|---|---|---|---|---|
| | AUDC | % Improvement | AUDC | % Improvement | AUDC | % Improvement |
| 3 | 8841.60(41.58) | 1.10(0.49) | 9305.36(74.03) | -4.08(0.44) | 9302.72(75.13) | -4.05(0.42) |
| 11 | 8905.19(31.78) | 0.39(0.60) | 9260.72(64.91) | -3.58(0.27) | 9264.40(82.37) | -3.62(0.32) |
| 18 | 8912.97(52.07) | 0.31(0.43) | 9220.87(94.56) | -3.13(0.36) | 9233.84(104.09) | -3.28(0.51) |
| 26 | 8939.26(61.24) | 0.01(0.07) | 9188.04(81.42) | -2.77(0.48) | 9210.00(88.54) | -3.01(0.48) |
| 34 | 8922.46(62.82) | 0.20(0.52) | 9099.73(78.70) | -1.78(0.41) | 9095.38(93.86) | -1.73(0.50) |
| 42 | 8940.63(65.05) | -0.00(0.00) | 9103.45(85.30) | -1.82(0.61) | 9108.91(79.07) | -1.88(0.37) |
| 50 | 8904.96(52.85) | 0.40(0.37) | 9100.83(79.09) | -1.79(0.59) | 9110.16(94.18) | -1.90(0.59) |
| 57 | 8909.19(58.59) | 0.35(0.35) | 9109.90(79.92) | -1.90(0.63) | 9113.88(82.44) | -1.94(0.48) |
| 65 | 8911.51(67.96) | 0.32(0.47) | 9103.57(56.82) | -1.83(0.45) | 9134.61(76.00) | -2.17(0.44) |

Table 6: AUDC scores and Percent improvement over the random baseline in XSUM of various sizes of $\mathcal{J}$ with standard deviations in brackets

| Size of $\mathcal{J}$ | OneTimeModel (Ours) | | OneTimeSoftmax | | OneTimeEntropy | |
|---|---|---|---|---|---|---|
| | AUDC | % Improvement | AUDC | % Improvement | AUDC | % Improvement |
| 3 | 8899.92(2045.15) | 30.52(5.80) | 10890.91(2394.00) | 14.94(6.98) | 12474.59(3613.93) | 4.39(6.83) |
| 5 | 9171.54(1900.72) | 28.23(4.67) | 10498.97(2180.89) | 17.76(7.69) | 12324.10(3610.31) | 5.59(7.60) |
| 7 | 9161.66(1952.92) | 28.37(4.83) | 10619.23(2300.14) | 16.91(7.95) | 12373.30(3584.54) | 5.12(7.84) |
| 9 | 9104.90(2044.80) | 28.96(4.65) | 10325.61(2141.19) | 19.06(8.20) | 12338.06(3644.00) | 5.50(7.83) |
| 11 | 9016.07(1924.03) | 29.48(5.00) | 10244.78(2072.60) | 19.62(8.15) | 12244.10(3588.08) | 6.18(7.62) |
| 13 | 9061.37(1970.45) | 29.14(5.37) | 10533.47(2169.80) | 17.44(8.35) | 12467.94(3755.22) | 4.64(8.86) |

Table 7: AUDC scores and percent improvement over the random baseline in MWP for various sizes of $\mathcal{J}$ with standard deviations in brackets (best training cost per $|J|$ for OneTimeModel).

as they efficiently maximize accuracy without requiring the large number of trainable parameters typically needed for $K$ separate binary classifiers.

Since the inputs of recurrent classifier chains at training time are typically ground truth labels, training token-level rejectors would similarly require using the correct rejection decision, $\mathbf{1}_{l\left(y,\widehat{y}_j^h\right)>c_j(x,\widehat{y}_{<j},y)}$, to determine the $j^{\text{th}}$ token prediction for the next label rejector. Following this strategy, also called "teacher forcing" Williams & Zipser (1989), can often cause a distribution shift, as label predictions will be used at test time instead of the correct label, resulting in errors propagating through the chain. Scheduled sampling Bengio et al. (2015) offers a middle ground by labeling the training data with the correct rejection decision with probability $p$, and otherwise using the predicted one. This probability $p$ often decays at a linear or exponential rate over epochs, gradually reducing the model's dependence on ground-truth rejection rules.

We use both Multi Layer Perceptrons (MLP) and Long Short-term memory (LSTM) models to perform token-level deferrals on XSUM data. All models are trained with teacher forcing, without teacher forcing, or with scheduled sampling. The Scheduled sampler decayed the teacher forcing probability exponentially by 0.95 with every epoch and the decay process stopped when the teacher forcing probability was 0.5. The sampler has 5 warmup epochs where the teacher forcing probability was 1.0 before the decay started. The coin flip of whether or not to teacher force occurred on a token-level, as opposed to a batch-level. Figure 4 shows the superiority of recurrent models like LSTMs over MLPs with teacher forcing adversely affecting their performance. The selected model is an LSTM trained without teacher forcing. We present the AUDC scores for each of the models in Table 8.

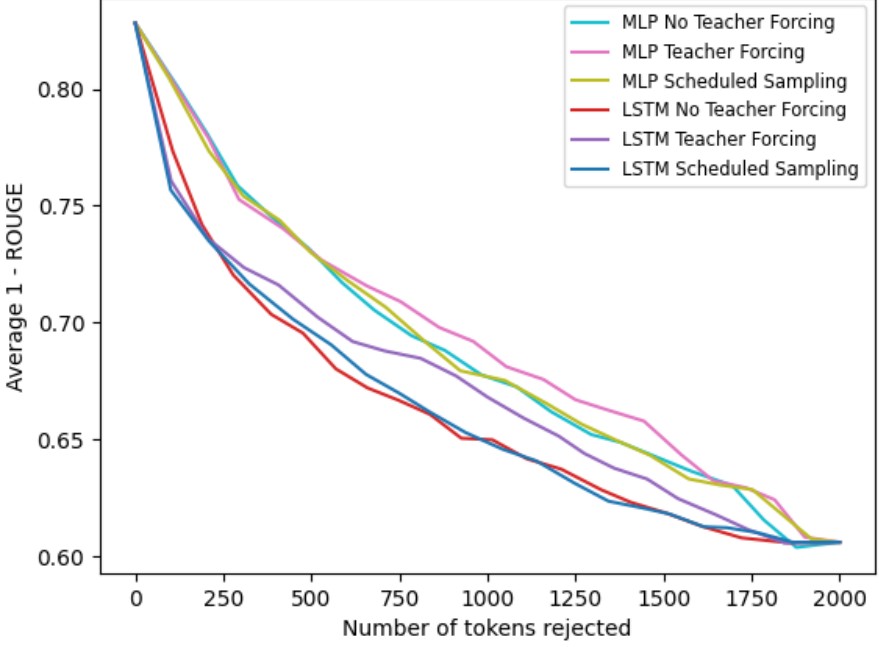

Figure 4: Cost-Loss plots for XSUM for various model architectures and training strategies.

## I  Broader Impacts Statement

This work proposes methods for learning to partially defer to a stronger expert for sequence prediction, with the goal of improving cost effectiveness of prediction systems. Such methods may be beneficial in scenarios

| Method | AUC Score | % Improvement |
|---|---|---|
| MLP No Teacher Forcing | 1375.48 (34.57) | 4.07 (1.01) |
| MLP Teacher Forcing | 1387.49 (25.96) | 3.22 (1.25) |
| MLP Scheduled Sampling | 1377.01 (31.92) | 3.96 (1.20) |
| LSTM No Teacher Forcing | 1323.92 (25.34) | 7.66 (0.96) |
| LSTM Teacher Forcing | 1340.43 (20.24) | 6.49 (1.24) |
| LSTM Scheduled Sampling | 1324.83 (27.04) | 7.60 (0.98) |
| TokenwiseSoftmax | 1354.78 (31.87) | 5.51 (0.94) |
| TokenwiseEntropy | 1360.41 (31.50) | 5.12 (0.62) |

Table 8: AUC Scores for various Token-level Rejector architectures and training

where expert resources are limited, allowing expert computation or human effort to be allocated to the portions of a prediction that are expected to provide the greatest benefit.

At the same time, partial-deferral systems should not be viewed as a replacement for human oversight. Miscalibrated routing decisions may fail to defer when expert intervention is needed, potentially degrading system performance or user trust. Consequently, deployment should include careful validation of the deferral decision, domain-specific calibration, and appropriate human oversight where errors carry significant consequences.

