# OpenReview forum: "Learning to Partially Defer for Sequences"
_TMLR — Under review for TMLR_

### Review · Reviewer_MtmY · 2026-06-28

**Summary Of Contributions:**

This paper studies the Learning to Defer (L2D) framework for sequence outputs, where a predictor may hand off part of its prediction to an expert instead of either predicting the full sequence itself or deferring the entire sequence at once. The main motivation is that deferral based on the entire sequence can be unnecessarily expensive for long outputs, especially when a subset of the predicted tokens or suffixes are unreliable. The paper proposes a more fine-grained formulation of deferral for sequences and develops post-hoc rejectors for pretrained sequence predictors. Overall, the reviewer thinks that the main contributions of this paper can be summarized from the following aspects: (1) extended L2D from standard multiclass/regression settings to sequential outputs and distinguished two practically relevant forms of granular deferral; (2) defined a sequence-level deferral loss by averaging tokenwise losses and expert costs over the output positions; (3) provided empirical results to justify the theoretical findings above.

**Additional Comments:**

NA

**Audience:**

Yes

**Audience Explanation:**

To the best of the reviewer’s knowledge, trustworthy ML, human-AI or expert-AI collaboration, selective prediction, and learning to defer are all active research topics nowadays. On the other hand, sequence models like transformers and modern LLMs are central to many high-impact applications in recent AI advancements. Therefore, extending L2D from scalar or multiclass outputs to structured sequential outputs is a relevant and timely contribution.

**Claims And Evidence:**

Yes

**Claims Explanation:**

From the reviewer's perspective, below are the major claims made in this paper:

(1) Partial deferral for sequences is a meaningful extension of L2D;

(2) The proposed surrogate losses are theoretically justified;

(3) Model-based partial rejectors improve cost-accuracy trade-offs over whole-sequence deferral and confidence-based alternatives.

Overall, the authors have supported the three major claims above both theoretically and empirically.

**Requested Changes:**

Overall, the reviewer is almost satisfied with the current manuscript. However, to further enhance the quality of the manuscript, the authors may consider the following changes in the revised manuscript:

(1) Clarify the gap between the theoretical training setup and deployment;

(2) Strengthen the discussion and analysis of the candidate set in one-time deferral;

(3) Add stronger ablations or analyses of the empirical gains.

---

> ### Author Response · Authors · 2026-07-08
>
> We thank the reviewer for the positive assessment of our work and for recognizing both the theoretical and empirical contributions. We appreciate the constructive suggestions and have addressed each point in the revised manuscript.
>
> > Comment: Clarify the gap between the theoretical training setup and deployment;
>
> Response: We have clarified this distinction in Section 3. We now explicitly state that the theoretical guarantees apply to the teacher-forced training protocol analyzed in the paper, whereas our strongest empirical results are obtained without teacher forcing to mitigate exposure bias. We also note that extending the theoretical analysis to autoregressive training remains an important direction for future work.
>
> > Comment: Strengthen the discussion and analysis of the candidate set in one-time deferral;
>
> Response: We have improved the discussion of the candidate handoff set in the main text (Section 6.5). We now discuss the tradeoff between routing flexibility and multiclass routing complexity, explaining why allowing all (L+1) handoff positions is not always the most cost-effective choice. We also added discussion in Section 6.4 on the role of the expert cost parameter in controlling the cost-quality tradeoff and referenced the corresponding sensitivity analyses.
>
> > Comment: Add stronger ablations or analyses of the empirical gains.
>
> Response: We strengthened the empirical evaluation in the updated submission. For XSUM, we removed the previous 15-word training filter, increased the maximum generated summary length from 20 to 64 tokens, expanded the evaluation to 200 test instances over 10 independent runs, and added other language-model cascade baselines (WholeModelScore). We also expanded the TSP and MWP evaluations to 200 test instances over 10 runs, with a longer prediction horizon for MWP. These changes provide a more comprehensive empirical validation while preserving the central conclusions of the paper.

---

### Review · Reviewer_eT88 · 2026-06-29

**Summary Of Contributions:**

The paper extends the Learning-to-Defer (L2D) framework from whole-prediction deferral to **partial** deferral for sequence outputs, so the system can interleave machine and expert along a sequence instead of deferring all-or-nothing. It introduces two post-hoc rejectors for a fixed pre-trained predictor, keyed to expert capability: a **token-level rejector** (defers individual tokens to a next-token-capable expert) and a **one-time rejector** (picks a single handoff point from which an expert completes the sequence, for non-autoregressive experts such as a Gurobi TSP solver). For each it gives a convex surrogate loss, a Bayes-consistency theorem, and Rademacher generalization bounds; one-time deferral is generalized to a candidate set J of handoff positions. Empirically, on TSP, XSUM summarization, and multi-step weather, it reports AUDC (area under the deferral curve) and % improvement over a random rejector, arguing partial deferral beats whole-sequence deferral and that model-based rejectors beat confidence-based ones at the same granularity.

## Key strengths:
1. Clean token-level vs one-time decomposition that maps to real expert constraints (the Gurobi "partial subtour as a solver constraint" example concretely motivates one-time deferral).
2. Complete, self-contained theory — convex surrogates, Bayes consistency, and finite-sample generalization bounds with explicit constants, with full appendix proofs.
3. The central qualitative finding holds across three diverse modalities and two baseline families, with honest reporting of where it does not help (modest TSP one-time gain).

## Key weaknesses:
1. One-time deferral is one of the paper's two proposed methods (OneTimeModel), yet its advantage is statistically significant only on TSP. On XSUM and MWP the OneTimeModel improvement is within one std (2.35±4.03, 7.23±11.80; n=5, no significance test) — both are bolded and §6.4 calls them "markedly better," so half the flagship contribution is under-supported on 2 of 3 datasets. The abstract's three-task framing compounds this (TSP is one-time-only and modest).
2. A model-based whole-sequence baseline (WholeModelEmbed) scores below the random rejector on all three tasks, so the model-vs-model comparison rests on an apparently non-functional baseline.
3. The cost-accuracy claim rides an abstract per-token deferral cost α_j; the real serving cost — re-prefill, and whether the predictor's KV cache can be reused by the expert — is never modeled, and decoder-only LLMs are sidestepped precisely because they would require reconstructing cross-model KV caches.

**Audience:**

Yes

**Audience Explanation:**

Learning-to-defer, selective prediction, and model cascades / LLM routing are active topics with a clear readership at TMLR. This paper extends consistent-surrogate L2D from single-label outputs to full sequences, with two deferral granularities and theoretical guarantees — a natural and useful question for those readers,

**Broader Impact Concerns:**

No or very minor concerns.

**Claims And Evidence:**

Yes

**Claims Explanation:**

The theoretical contributions (convex surrogates, Bayes consistency, generalization bounds) are stated with explicit assumptions and full appendix proofs, and read as sound. The strongest empirical claims are well above noise — token-level deferral on XSUM (+7.63±0.93%) and MWP (+25.78±1.33%), and one-time deferral on TSP (+11.52±2.41%) — so the central direction (partial ≥ whole; model-based ≥ confidence-based) is supported.

Three things keep the evidence short of fully convincing as currently stated: (1) one-time deferral — one of the two proposed methods — is convincingly supported only on TSP (11.52±2.41); on XSUM and MWP the OneTimeModel gains are within one std (2.35±4.03; 7.23±11.80, n=5, no significance test) yet are bolded and called "markedly better," and the MWP column is so noisy (AUDC std ≈150 on means ≈500) that methods are barely separable there; (2) the model-based whole baseline WholeModelEmbed is below random on every task (−0.58 / −0.11 / −5.28), so "outperforms model-based whole baselines" is not convincingly shown; and (3) the reported "cost" is only an abstract per-token count, not a real serving cost.

On the third point: the headline result is a cost-accuracy tradeoff, but the paper measures "cost" only as α_j, which simply grows with the number of deferred tokens. It never measures the actual compute or latency. This matters most for the small-model / large-model use case. When the small model hands off to the large model, does the large model reuse the small model's KV cache, or must the input be re-prefilled? The paper does not say. §6.1 even notes that decoder-only models would have to reconstruct cross-model KV caches, "which defeats the purpose of partial feedback," and the authors avoid this only by using an encoder-decoder model (T5). So once a realistic serving cost is counted, the benefit could shrink — and it may not carry over to decoder-only LLMs at all.

**Requested Changes:**

## Critical:
1. Hedge the one-time XSUM/MWP claims: OneTimeModel — one of the two proposed methods — is significant only on TSP, while on XSUM (2.35±4.03) and MWP (7.23±11.80) its advantage is within one std with no significance test, yet both are bolded and called "markedly better." Either soften to within-noise language or add a paired significance test on the existing 5×100 runs, and scope the abstract's three-task framing (TSP is one-time-only and modest). Token-level and one-time-TSP claims can stay strong.
2. Clarify the WholeModelEmbed baseline — show it was trained/tuned comparably so it is a credible model-based whole-sequence rejector, or temper the "outperform model-based whole-sequence baselines" claim.

## Would-strengthen:
3. Account for the real cost, and clarify the decoder-only story. §6.1 declines decoder-only LLMs because they "require reconstructing cross-model KV caches," implying encoder-decoder T5 avoids this. But T5 still cannot reuse the other model's KV across different weights — on every handoff the resuming model must re-prefill the other model's inserted tokens to rebuild its own decoder self-attention K/V (the encoder/cross-attention is per-model and reused; the self-attention for the handed-over span is not). Please clarify: (a) how exactly does encoder-decoder avoid the "reconstruction" decoder-only allegedly needs, given the same cross-model non-reusability applies to both? (b) Is decoder-only merely extra cost (re-prefilling the handoff span) or genuinely infeasible — since decoder-only LLMs now dominate, this decides the method's relevance. (c) Report actual compute/latency including this re-prefill (token-level interleaving pays it on every handoff), not just the rejected-token count, and whether the cost-accuracy benefit survives.
4. Relabel Figure 1's y-axis: it plots a loss (1 − ROUGE-1, lower-is-better) but is titled "Average 1-ROUGE Score," which both calls a loss a "Score" and visually collides with the metric name ROUGE-1 — easy to read backwards. Use e.g. "Average (1 − ROUGE-1) ↓", add axis ranges/legends, and bring the TSP/MWP deferral curves into the main text (only XSUM is shown).
5. Lower the per-pair adaptation cost / consider a lighter or training-free rejector. Because the rejector consumes the predictor's encoder/decoder hidden states, it is tied to one specific predictor and must be retrained for every new (predictor, expert, task) triple (XSUM token-level ≈5h; one-time needs an offline "mixed-prediction" dataset), and obtaining labels requires running the expert. For deployment across many model pairs this adaptation cost is a real barrier. Could a lighter-weight or training-free rejector work — e.g. a prompt-based LLM judge, or a feature-light / transferable rejector that generalizes across predictor–expert pairs without per-pair retraining — and how would it compare to the trained model-based rejector? (The paper compares only training-free confidence rejectors, which still read that one predictor and do not transfer across pairs.)

---

> ### Author Response · Authors · 2026-07-08
>
> We thank the reviewer for their evaluation and for recognizing the theoretical contributions of the paper. We particularly appreciate the suggestions regarding the empirical evaluation and practical deployment considerations. In response, we expanded the experiments, clarified the implementation details for encoder-decoder models, strengthened the description of the baselines, and moderated the claims where the empirical evidence isn’t as strong.
>
> >  Comment: Hedge the one-time XSUM/MWP claims …
>
> Response: In response to Reviewer qkYS, we substantially expanded the empirical evaluation for all three datasets in the revised manuscript. Specifically, we increased the evaluation from 100 to 200 test instances and now report averages over 10 independent runs (previously 5). For MWP, we also increased the prediction horizon from 6 to 12 steps. These changes significantly reduce the variance of the estimates. In the revised experiments, OneTimeModel continues to demonstrate statistically meaningful improvements on both TSP (7.98 ± 1.72) and MWP (30.76 ± 4.86). For XSUM, the gains remain modest (1.10 ± 1.06), and we have revised the discussion to explicitly acknowledge this rather than characterizing the improvement as substantial. We have also included one-sided paired significance tests in Appendix E comparing the random rejector with the other baselines. Both our one-time and token-level models have p-values less than 0.05.
>
> > Comment: Clarify the WholeModelEmbed baseline
>
> Response: In the revised manuscript, we expanded Appendix C to provide a detailed description of all whole-sequence baselines, including WholeModelEmbed, together with their training procedures and hyperparameter settings. This clarifies that the baseline was trained under the same experimental protocol as the proposed rejectors.
>
> > Comment: Account for the real cost, and clarify the decoder-only story.
>
> Response: We have revised Section 6.1 (under XSUM) to clarify the implementation details. Specifically, when switching between predictor and expert, the expert recomputes both its encoder and decoder states. The advantage of the encoder-decoder architecture is instead that the source article is encoded once per model, whereas in decoder-only architectures the prompt forms part of the autoregressive context and typically requires replaying the prompt together with the generated prefix after each handoff, increasing the computational overhead of frequent token-level deferrals. We have also expanded the discussion section (Section 7) to describe extending the framework to decoder-only models and reducing this overhead through future architectural improvements.
>
> > Comment: Relabel Figure 1's y-axis:
>
> Response: We thank the reviewer for this suggestion. We have relabeled the y-axis to "Average (1 − ROUGE-1) ↓" and brought the TSP and MWP deferral curves to the main text.
>
> > Comment: Lower the per-pair adaptation cost…
>
> Response: We agree that the need to retrain the rejector for each predictor-expert-task combination may limit the practicality of deployment. In response, we have expanded the discussion section to highlight this as an important direction for future work.

---

### Review · Reviewer_qkYS · 2026-07-02

**Summary Of Contributions:**

This paper extends the learning-to-defer (L2D) framework to sequence-output prediction by proposing two partial-deferral settings: token-level deferral, where a rejector decides at each position whether to use the predictor or a next-token expert, and one-time deferral, where the rejector selects a single handoff point after which the expert completes the remaining suffix. The paper derives convex surrogate losses for both settings, proves Bayes consistency and generalization bounds under fixed-predictor assumptions, and evaluates the methods on three tasks: Traveling Salesman Problem solving, XSUM news summarization, and multi-step weather prediction.

- Key strengths: The problem formulation is clean and the theoretical analysis is more complete than typical applied ML papers, covering surrogate consistency and finite-sample generalization bounds. The distinction between token-level and one-time deferral is a useful conceptual contribution.

- Key weaknesses: The empirical evaluation is small-scale and limited in scope. The only NLP experiment uses short XSUM summaries with T5-small/T5-base, which is far from modern LLM deployment. Expert cost models are simplified proxies rather than realistic costs. The gains over baselines are modest in several settings, and the strongest related LM cascade baselines are not fully compared.

**Additional Comments:**

I recognize that the theoretical contribution is the primary value of this paper, and the surrogate analysis and generalization bounds are solid. However, the empirical evidence is not strong enough to support the paper's broader claims about practical improvements in cost-quality tradeoffs for sequence prediction. The experiments feel like a proof-of-concept rather than a convincing empirical validation. For TMLR, I would expect either stronger empirical evidence or a more explicitly theory-focused framing that does not rely on the current experiments to carry the practical claims.

**Audience:**

Yes

**Audience Explanation:**

The partial-deferral formulation for sequence outputs is a natural and underexplored extension of L2D. Researchers working on learning to defer, selective prediction, cost-sensitive learning, model cascades, and human-AI collaboration would find the theoretical framework interesting.

**Broader Impact Concerns:**

The paper does not include a dedicated broader impact statement. While the work is primarily theoretical, partial deferral systems could be deployed in high-stakes settings where incorrect routing decisions have real consequences. I recommend that the authors add a brief statement acknowledging that automated deferral routing should not replace human accountability or domain-specific validation, and that miscalibrated rejectors could lead to systematic over-reliance on automated predictions in cases where expert consultation is warranted.

**Claims And Evidence:**

No

**Claims Explanation:**

The theoretical claims are generally sound, but the empirical evidence is not convincing enough to support the broader claims made in the paper. Specifically:

First, the experiments are small-scale. Each task uses only 100 test instances per run with five runs. For XSUM, this is insufficient to draw reliable conclusions about ROUGE-1 or AUDC differences in the 1–3% range.

Second, the XSUM experiment is severely constrained. The token-level rejector is trained only on summaries shorter than 15 words, and summaries are capped at 20 tokens. This does not reflect realistic summarization settings and limits the conclusions about token-level deferral for language generation.

Third, the expert cost models are simplified proxies. For XSUM, the token-level expert cost is whether the expert token appears in the reference, not a real API or human cost. For weather prediction, the expert is a simulated Gaussian-noise oracle. For one-time deferral, the cost schedule alpha_j is hand-designed and the results are sensitive to it.

Fourth, the paper does not compare against the strongest related baselines. Language Model Cascades: Token-level Uncertainty and Beyond directly studies learned post-hoc deferral rules using token-level uncertainty and intermediate embeddings for generative LMs. The paper only includes a WholeModelEmbed variant from that work, not the stronger learned quantile or embedding-based cascade methods, which are the most relevant comparison for the XSUM experiment.

Fifth, the theoretical analysis assumes a fixed predictor and teacher-forced contexts, while the best empirical token-level rejector is trained without teacher forcing. This gap between theory and practice is not adequately explained.

**Requested Changes:**

# Critical for acceptance:

- The empirical evaluation must be strengthened. The current experiments are too small and too constrained to support the paper's claims. At minimum, the XSUM experiment should be run on a full test set without the 15-word training filter, and the test set size should be substantially larger than 100 instances per run.

- The paper must compare against the strongest LM cascade baselines for the XSUM experiment, specifically the learned quantile-based and intermediate-embedding-based deferral rules from Language Model Cascades: Token-level Uncertainty and Beyond. Without this comparison, it is unclear whether the proposed model-based token-level rejector offers a meaningful advantage over existing learned cascade methods.

- The cost model must be made more realistic or the claims must be scoped accordingly. Using token correctness as expert cost for XSUM and a Gaussian-noise oracle as the expert for MWP does not reflect real deployment. If the paper cannot use realistic costs, it must clearly state that the results are valid only under these specific proxy cost assumptions.

- The gap between theory and the best empirical setup must be explicitly addressed. The theoretical analysis assumes teacher-forced contexts, but the best XSUM rejector is trained without teacher forcing. The paper should either provide theoretical support for the no-teacher-forcing regime or clearly state that the theory covers a different training protocol than the one used in the best empirical result.

- The sensitivity of one-time deferral to the cost schedule and the candidate handoff set must be discussed in the main text, not only in the appendix. The current appendix results show that performance is sensitive to alpha_j and that the full set J=L+1 is not optimal for XSUM. This is important for understanding the robustness of the method.

# Would strengthen the work:

- Add a modern decoder-only LLM experiment, or at minimum provide a thorough discussion of why token-level deferral is not straightforward for decoder-only models and what would be needed to extend the framework.

- Add simpler baselines for one-time deferral, such as deferring at the first low-confidence token, cumulative uncertainty thresholding, and random handoff matched by cost budget.

- Report results under real or proxy deployment costs such as expert API calls, total generated tokens, and latency, rather than only task-specific proxy losses.

- Verify the notation in Theorem 4.1. The consistency condition compares costs using what appears to be the same tilde c_i term for both positions i and j, which may be a typo or notation inconsistency.

---

> ### Author Response · Authors · 2026-07-08
>
> We thank the reviewer for the constructive feedback. We agree that strengthening the empirical evaluation and clarifying several aspects of the methodology would improve the paper. In the revised manuscript, we have expanded the XSUM experiments, added the suggested language model cascade baselines, clarified the scope of the theoretical guarantees, and discussed some of the ablations in the main text.
>
> > Comment: The empirical evaluation must be strengthened...
>
> Response: Thank you for this suggestion. We have substantially expanded the scope of XSUM evaluation in the revised manuscript. Specifically:
> - We removed the previous 15-word training filter.
> - We increased the maximum generated summary length from 20 to 64 tokens, making the task considerably more representative of practical summarization settings.
> - We increased the evaluation from 100 to 200 test instances, and report averages over 10 independent runs (previously 5).
>
> These changes strengthen the empirical evaluation while preserving the conclusions drawn from the older results.
>
> > Comment: The paper must compare against the strongest LM cascade baselines...
>
> Response: In the revised manuscript, we include both WholeModelScore and WholeModelEmbed, corresponding respectively to the PostHocQuantile and PostHocEmbed methods proposed in Gupta et al. These additions provide direct comparisons against the strongest learned post-hoc routing baselines considered in that work.
>
> > Comment: The cost model must be made more realistic ...
>
> Response: We agree that our experiments use expert correctness rather than deployment-specific costs such as API latency or monetary cost. This distinction is intentional. Quantities such as latency or API cost are not directly comparable to the predictor's error. For example, comparing a token classification error with a dollar cost is not an apples-to-apples comparison. Instead, our framework separates these two considerations. The rejector is trained to estimate when expert intervention is likely to improve prediction quality, while deployment-specific costs are naturally incorporated through the expert budget. In practice, a user can select the rejector threshold to satisfy a desired budget (e.g., a limit on API calls, latency, or monetary expenditure), thereby controlling how frequently the expert is queried. Under this fixed budget, the objective becomes allocating expert intervention to the examples or tokens where it yields the greatest improvement in prediction quality.
>
> > Comment: The gap between theory and the best empirical setup must be explicitly addressed...
>
> Response: We now explicitly distinguish the theoretical and empirical training protocols in Section 3. We state there that the theoretical guarantees apply to teacher-forced training, whereas the strongest empirical XSUM results are obtained without teacher forcing to mitigate exposure bias. We further note that extending the analysis to autoregressive training remains an important direction for future work.
>
> > Comment: The sensitivity of one-time deferral to the cost schedule…
>
> Response: Following the reviewer’s suggestion, we have included the discussion of both the expert cost (Section 6.4) and size of $\mathcal{J}$  (Section 6.5) ablation into the main text.
>
> > Comment: Add a modern decoder-only LLM experiment
>
> Response: We agree that evaluating modern decoder-only LLMs is an important direction. Since our primary contribution is the formulation and analysis of partial deferral, we instead expanded the discussion to clarify the computational considerations for decoder-only architectures (Section 6.1 under XSUM) and outline possible implementation strategies for future work (Section 7).
>
> > Comment: Verify the notation in Theorem 4.1.
>
> Response: Typo has been fixed.
>
> > Comment: The paper does not include a dedicated broader impact statement.
>
> Response: Thank you for bringing this to our attention. We have now included a broader impacts statement in Appendix I.

---

> > ### Comment · Reviewer_qkYS · 2026-07-17
> > **Response**
> >
> > Thank you for the detailed response and for making substantial revisions. I appreciate that the revised manuscript appears to address several of my main concerns: the XSUM setup has been expanded by removing the short-summary filter, increasing the maximum generation length, using more test instances and runs, and adding stronger post-hoc cascade baselines; the theory/empirical-protocol gap is now explicitly scoped; and the cost-schedule and candidate-handoff-set discussions have been moved into the main text.
> >
> > Before I finalize my recommendation, I would like to ask for clarification on a few remaining points.
> >
> > First, regarding the XSUM evaluation: the revised setup is clearly stronger than the original one, but it still evaluates on 200 test instances rather than the full test set. Could you clarify why 200 instances were chosen, and whether the main conclusions are stable under a larger evaluation set or the full XSUM test set? If a full evaluation is computationally prohibitive, it would be helpful for the final manuscript to state this limitation clearly.
> >
> > Second, regarding the language-model cascade baselines: you state that WholeModelScore and WholeModelEmbed correspond to PostHocQuantile and PostHocEmbed from Gupta et al. Could you clarify whether these baselines are tuned under the same validation protocol and budget grid as the proposed rejectors? Since the strength of the empirical claim depends on these comparisons, I think the final version should make the equivalence to the prior cascade methods and the tuning procedure very explicit.
> >
> > Third, and most importantly, regarding the cost interpretation: I understand your point that deployment-specific quantities such as latency, API cost, or monetary cost can be controlled through a budget/threshold. However, the current experiments still evaluate cost using proxy expert-budget quantities rather than actual serving cost. In token-level deferral, realistic serving cost may depend strongly on handoff frequency, recomputation/prefill overhead, and whether the expert needs to replay the generated prefix. Your clarification about encoder-decoder versus decoder-only architectures is helpful, but I think the final manuscript should explicitly state that the reported cost-quality curves should be interpreted under the paper’s proxy cost model, not as evidence of end-to-end latency or monetary savings in deployed LLM systems.
> >
> > Relatedly, for decoder-only LLMs, the current response frames this as future work. I think this is reasonable, but I would encourage the final manuscript to avoid implying that the empirical results directly establish practical benefits for modern decoder-only LLM deployments.
> >
> > Overall, the revisions substantially mitigate my original concerns, especially on the empirical scale, baseline coverage, and theoretical scoping. My remaining concern is mainly about claim calibration: the paper is strongest as a theoretically grounded formulation of partial deferral for sequences with empirical validation under proxy cost assumptions. It would be less convincing if framed as demonstrating practical deployment cost savings without measuring realistic compute/latency/API costs.

---

> > > ### Author Response · Authors · 2026-07-22
> > >
> > > Thank you for your follow-up comments. We appreciate your suggestions regarding claim calibration and have revised the manuscript accordingly.
> > >
> > > - XSUM evaluation size: We have clarified in the Evaluation subsection that XSUM is evaluated on a random subset of 2,000 test instances to keep repeated evaluation across ten runs and multiple rejector baselines computationally tractable. We also note that, although evaluating on the full test set would be more comprehensive, the low variability across runs suggests that the qualitative conclusions are unlikely to change.
> > > - WholeModelScore / WholeModelEmbed baselines. We now explicitly state in the Baselines subsection that WholeModelScore and WholeModelEmbed are our implementations of the PostHocQuantile and PostHocEmbed methods of Gupta et al., respectively, making the correspondence with the prior work explicit. As in our experiments, all model-based rejectors (both baselines and our proposed methods) were trained using the same training dataset and evaluated over the same threshold grid to ensure a fair comparison.
> > > - Interpretation of the cost-quality curves: Following your suggestion, we have added an explicit statement in the Evaluation subsection clarifying that the reported deferral curves should be interpreted under the task-specific cost models defined for each experiment, and should not be viewed as measurements of deployment-specific quantities such as inference latency.
> > > - Decoder-only LLMs: We have also revised the Discussion section to better calibrate our claims regarding decoder-only models. The manuscript now emphasizes that our language-model experiments validate the proposed partial-deferral framework in the encoder-decoder setting and should not be interpreted as demonstrating similar deployment benefits for modern decoder-only LLM systems, whose practical deployment requires accounting for architecture-specific inference costs.
> > >
> > > We thank the reviewer again for these suggestions.